# INTERLEAVE-VLA: ENHANCING ROBOT MANIPULATION WITH INTERLEAVED IMAGE-TEXT INSTRUCTIONS

**Cunxin Fan**[1*]     **Xiaosong Jia**[2*]     **Yihang Sun**[1]     **Yixiao Wang**[3]     **Jianglan Wei**[3]

**Ziyang Gong**[1]     **Xiangyu Zhao**[1]     **Masayoshi Tomizuka**[3]     **Xue Yang**[1†]     **Junchi Yan**[1†]

**Mingyu Ding**[4†]

[1]SCS & SAIS & SAI & ICISEE, Shanghai Jiao Tong University
[2]Institute of Trustworthy Embodied AI, Fudan University
[3]UC Berkeley     [4]UNC-Chapel Hill     *Equal Contributions     †Correspondence Authors

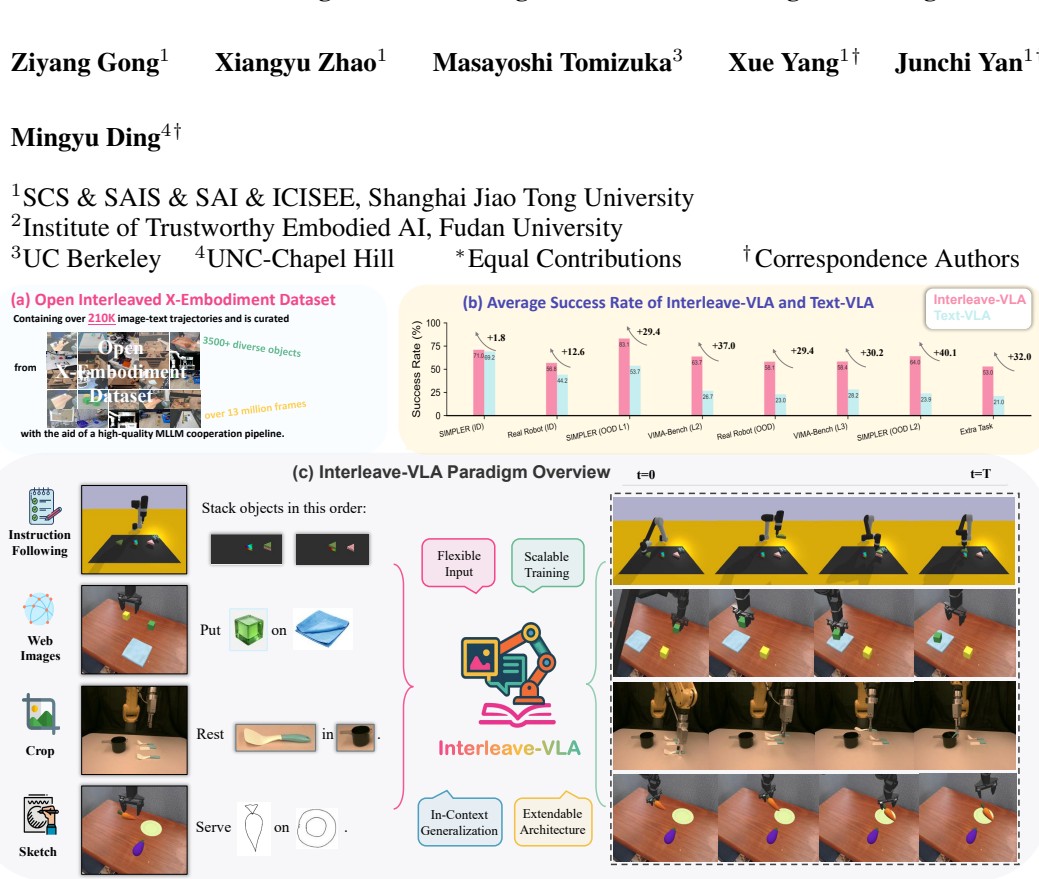

Figure 1: **(a)** Our Interleaved X-Embodiment Dataset features diverse, high-quality object-centric images automatically generated from real-world robot demonstrations. **(b)** Interleave-VLA achieves $2\times$ stronger out-of-domain generalization compared to text-only VLA models in both simulation and real-robot experiments. **(c)** It enables flexible, zero-shot instruction following with cropped images, web photos, and hand-drawn sketches for practical and intuitive human-robot interaction.

## ABSTRACT

The rise of foundation models paves the way for generalist robot policies in the physical world. Existing methods relying on text-only instructions often struggle to generalize to unseen scenarios. We argue that interleaved image-text inputs offer richer and less biased context and enable robots to better handle unseen tasks with in-context visual grounding. Building on this insight, we introduce Interleave-VLA, a robot learning paradigm extending interleaved image-text instructions from digital world to directly generating continuous action sequences in the physical world. It offers a natural, flexible, and model-agnostic paradigm that extends state-of-the-art vision-language-action (VLA) models with minimal modifications while achieving strong zero-shot generalization. Interleave-VLA also includes an automatic pipeline that converts text instructions from Open X-Embodiment into interleaved image-text instructions, resulting in a large-scale

†Emails: {yangxue-2019-sjtu, yanjunchi}@sjtu.edu.cn, md@cs.unc.edu

real-world interleaved embodied dataset with 210k episodes. Comprehensive evaluation in simulation and real world show that Interleave-VLA offers two major benefits: **(1)** improves out-of-domain generalization to unseen objects by $2\times$ compared to text input baselines, **(2)** supports flexible task interfaces and diverse instructions in a **zero-shot manner**, such as hand-drawn sketches. We attribute Interleave-VLA's strong zero-shot capability to the use of instruction images, which effectively mitigate hallucinations, and the inclusion of heterogeneous multimodal datasets, enriched with Internet-sourced images, offering potential for scalability. Our website has more information.

# 1 INTRODUCTION

The remarkable success of large language models (LLMs) (Achiam et al., 2023; Touvron et al., 2023; Bai et al., 2023; Liu et al., 2024a) and vision-language models (VLMs) (Bai et al., 2025; Team, 2024; Liu et al., 2023a; Chen et al., 2024a; Luo et al., 2025a) has established the paradigm of foundation models in the digital world, which are capable of generalizing across a wide range of tasks and domains. Inspired by this progress, the robotic community is actively developing robotic foundation models (Brohan et al., 2023; Kim et al., 2024; O'Neill et al., 2024; Black et al., 2024; Intelligence et al., 2025; Chi et al., 2023) to bring similar generalizability to unseen tasks and scenarios into the physically embodied world. Despite these advances, effective out-of-domain generalization of robotic policies remains a key challenge. We argue that the predominant reliance on text-only instructions in current generalist robotic policies constrains their ability to generalize. Text instructions often prove ambiguous or cumbersome in scenarios where users need to specify goals like "pick up an object like this," referring to a uniquely shaped or colored item. In contrast, interleaved image-text instructions allow robots to interpret unseen tasks more effectively by providing in-context visual and textual cues, beyond what text instructions alone can convey.

To develop a general and practical robot policy capable of acting on interleaved image-text instructions in the real world, a straightforward solution is to build upon VLA (Kim et al., 2024; O'Neill et al., 2024; Brohan et al., 2022; 2023; Black et al., 2024; Team et al., 2025) models, which naturally extend VLMs by incorporating action understanding and generation, making them well-suited for robotic tasks. However, current VLA models (Brohan et al., 2023; Kim et al., 2024; Black et al., 2024) remain predominantly trained on text-only instructions—a setting we refer to as the Text-VLA paradigm. This limits their ability to benefit from multimodal instruction signals, which have been shown to enhance generalization in vision-language learning (Achiam et al., 2023; Team et al., 2025). While VIMA (Jiang et al., 2023) served as a conceptual pioneer for multimodal robotics, it was restricted to high-level planning in stylized 2D simulations, without further exploring the practicality and generalizability of interleaved instructions. Therefore, there is an urgent need to systematically investigate the potential benefits of training and testing on interleaved instructions over the text inputs that virtually all modern VLAs still adhere to.

To address this limitation, we first build a high-quality interleaved image-text datasets, crucial for training multimodal models. In order to bridge the gap of the lack of image-text interleaved datasets in robotic manipulation, we develop a pipeline to automatically construct interleaved instructions from existing datasets. The proposed pipeline enables automatic and accurate generation of interleaved instructions from real-world dataset Open X-Embodiment (O'Neill et al., 2024). The resulting interleaved dataset contains over 210k episodes and 13 million frames, making it a large-scale, real-world interleaved embodied dataset. This enables training Interleave-VLA with real-world interaction data and diverse visual instruction types.

We then propose a new paradigm called Interleave-VLA, designed for generating continuous actions from interleaved inputs. As illustrated in Figure 2, Interleave-VLA consists of three key components: **(1)** a lightweight adaptation module that introduces special separator tokens into the tokenizer, enabling existing VLAs to process interleaved inputs without architectural changes, **(2)** a scalable training pipeline that leverages large-scale interleaved embodied datasets while preserving original objectives and hyperparameters, and **(3)** a versatile inference interface that supports both text-only and interleaved instructions, allowing the use of real-world camera crops, web images, or even sketches at test time. This effective design unlocks multimodal instruction-following capabilities for state-of-the-art VLAs in $\pi_0$ and can be readily extended to other VLA models, such as

OpenVLA (Kim et al., 2024). Experimental results demonstrate that Interleave-VLA significantly surpasses text-only baselines in out-of-domain tasks. The interleaved format enables robust zero-shot generalization to novel objects and user-provided sketches unseen during training.

We further investigate the factors behind Interleave-VLA's superior zero-shot performance relative to Text-VLA. We find that both the scale and heterogeneity of the Interleaved X-Embodiment Dataset contribute to consistent gains in both low- and high-data regimes. We also categorize three recurring forms of attentional hallucination in Text-VLA: *attentional bias*, *diffused attention*, and *attention leakage*, which arise from linguistic ambiguity and distributional biases in text-only instruction interpretation. We summarize our key takeaways below:

- Generalization failures in VLAs often stem from **attentional hallucinations**, which we summarized as attentional bias, diffused attention, and attention leakage, driven by **(1)** ambiguous contexts and **(2)** training distribution biases (Section 4.1).

- Interleaved image-text instructions mitigate the hallucination caused by **ambiguous contexts**, providing less-biased in-context visual grounding for better generalization (Figure 5 in Section 4.1).

- Modality diversity (e.g, interleaved data, web data) alleviates the hallucination from **training distribution biases**, further enhancing generalization. Cross-modal training benefits performance in both interleaved evaluation and text-only evaluation (Section 4.3.3).

Our core contribution is threefold. **(1)** We introduce Interleave-VLA: a lightweight, transferable paradigm that enhances the generalization capability of current text input VLA models with interleaved image-text instructions. Through comprehensive evaluation, we demonstrate **2×** gains in out-of-domain generalization to novel objects, along with emergent zero-shot capabilities for interpreting diverse visual instructions, such as **hand-drawn sketches**. **(2)** We opensource a large-scale, real-world interleaved embodied dataset with 210k episodes and 13 million frames based on Open X-Embodiment, generated by a fully automated pipeline. **(3)** We provide insights into Interleave-VLA's effectiveness in mitigating attentional hallucinations commonly observed in Text-VLA.

## 2 RELATED WORK

**Interleaved Vision-Language Models.** In the digital domain, recent advances in vision-language models have evolved from handling simple image-text pairs (Liu et al., 2023a; Radford et al., 2021; Li et al., 2023; Fang et al., 2023) to processing arbitrarily interleaved sequences of images and text (Bai et al., 2025; Team, 2024; Chen et al., 2024a; Luo et al., 2025a; Xue et al., 2024; Li et al.; Alayrac et al., 2022; Chen et al., 2024b; Jiang et al.). This interleaved format allows models to leverage large-scale multimodal web corpora—such as news articles and blogs—where images and text naturally appear in mixed sequences. Such models have demonstrated improved flexibility and generalization, enabling transfer across diverse tasks and modalities (Li et al.). Despite these successes in the digital world, robotic foundation models in the physical world have yet to fully exploit the benefits of interleaved image-text instructions. Motivated by the progress of interleaved VLMs, we extend this paradigm to the action modality, enabling vision-language-action models to process interleaved instructions. Our results show that multimodal learning with interleaved inputs greatly boosts generalization and displays emergent capabilities in robotic manipulation tasks.

**Vision Language Action Models.** Vision-language-action (VLA) models have advanced robotic manipulation by enabling policies conditioned on both visual observations and language instructions (Kim et al., 2024; Team et al., 2025; Brohan et al., 2022; 2023; Black et al., 2024; Fang et al., 2025a; Wen et al., 2025; Bjorck et al., 2025; Intelligence et al., 2025),. Most prior VLA models process single (Kim et al., 2024) or multiple (Brohan et al., 2023; Black et al., 2024) observation images with text-only instructions, with some exploring additional modalities such as 3D (Zhen et al., 2024) and audio (Zhao et al., 2025b). Only few existing works, such as VIMA (Jiang et al., 2023), explore the use of interleaved instructions in robotics, evaluating vision-language planning tasks within a high-level 2D action space in simulation. However, they have not investigated the broader benefits of interleaved image-text instructions, such as (1) their advantages over text-only instructions and (2) their applicability to real-world scenarios involving low-level robotic actions. As a result, the practical value of this paradigm remains underexplored due to a lack of real-world datasets and policies capable of handling such input In this work, we make the first step to bridge this gap by proposing Interleave-VLA: a simple, model-agnostic paradigm that extends existing VLA models to

| Method | Plug-in | Multimodal Instructions | Backbone Agnostic | No External Data Source | No Simulation / Physics Engine | Auto Data Augmentation | Customized Image Instruction |
|---|---|---|---|---|---|---|---|
| Gemini Robotics (Team et al., 2025) | ✗ | ✗ | ✓ | ✗ | ✓ | ✗ | ✗ |
| GR00T N1 (Bjorck et al., 2025) | ✗ | ✗ | ✗ | ✗ | ✗ | ✗ | ✗ |
| $\pi_{0.5}$ (Intelligence et al., 2025) | ✗ | ✗ | ✗ | ✗ | ✓ | ✓ | ✗ |
| CoT-VLA (Zhao et al., 2025a) | ✗ | ✗ | ✗ | ✓ | ✓ | ✓ | ✗ |
| Helix (Cui et al., 2025) | ✗ | ✗ | ✓ | ✗ | ✓ | ✓ | ✗ |
| ReBot (Fang et al., 2025b) | ✓ | ✗ | ✓ | ✓ | ✗ | ✓ | ✗ |
| VLA-0 (Goyal et al., 2025) | ✓ | ✗ | ✓ | ✓ | ✓ | ✗ | ✗ |
| Being-H0 (Luo et al., 2025b) | ✗ | ✗ | ✗ | ✗ | ✓ | ✓ | ✗ |
| VLAS (Zhao et al., 2025b) | ✗ | ✓ | ✗ | ✗ | ✓ | ✓ | ✗ |
| NaVILA (Cheng et al., 2025) | ✗ | ✗ | ✓ | ✗ | ✗ | ✓ | ✗ |
| NORA (Hung et al., 2025) | ✗ | ✗ | ✗ | ✓ | ✓ | ✗ | ✗ |
| **Interleave-VLA (Ours)** | ✓ | ✓ | ✓ | ✓ | ✓ | ✓ | ✓ |

Table 1: **Comparing Interleave-VLA with representative VLA methods.** Unlike prior systems that depend on fixed backbones, source external Internet or simulation data, and accept only text inputs, Interleave-VLA operates as a backbone-agnostic plug-in that supports image-text instructions. It reuses existing robot datasets without relying on external data acquisition, provides automatic, scalable data augmentation, and uniquely enables custom image-conditioned instructions (like sketches) at test time, yielding a more versatile, practical and generalizable VLA paradigm.

support interleaved image-text instructions with minimal modifications. Our comprehensive experiments demonstrate that interleaved instructions substantially improve generalization, and unlock strong zero-shot capabilities for diverse user-provided inputs. This highlights the practicality and scalability of interleaved image-text instructions for real-world robotic manipulation.

# 3 INTERLEAVE-VLA AND OPEN INTERLEAVED X-EMBODIMENT DATASET

## 3.1 PROBLEM FORMULATION

A discrepancy exists between the input modalities of modern Vision-Language Models (VLMs) (Alayrac et al., 2022; Bai et al., 2025; Team, 2024; Xue et al., 2024), which accept arbitrarily interleaved inputs, and most Vision-Language-Action (VLA) models, which typically operate on a single text instruction. We formally define this text-only instruction paradigm as **Text-VLA**. In this work, we propose **Interleave-VLA**, a generalized paradigm that allows a robotic policy to generate actions conditioned on interleaved image-text inputs. This formulation elevates VLA models to the same input flexibility as VLMs, thereby rendering Text-VLA a specialized instance.

Formally, a policy $\pi_\theta$ under the Interleave-VLA paradigm generates an action $a_t$ at each timestep $t$ by sampling from a distribution conditioned on the state $s_t$: $a_t \sim \pi_\theta(\cdot \mid s_t)$ where the state is defined as a tuple $s_t = (I_t, \mathbf{q}_t, \mathcal{I})$. This tuple comprises the current visual observation $I_t$ (e.g., an image or set of images), the robot's proprioceptive state $\mathbf{q}_t$, and an interleaved instruction sequence $\mathcal{I}$. The ordered instruction sequence is represented as $\mathcal{I} = (u_1, \ldots, u_M)$, where each token $u_j \in \mathcal{V}_{\text{text}} \cup \mathcal{V}_{\text{img}}$ belongs either to the set of text tokens $\mathcal{V}_{\text{text}}$ or to the set of image tokens $\mathcal{V}_{\text{img}}$. Notice that Interleave-VLA can degenerate to special case of standard Text-VLA when all $u_j \in \mathcal{V}_{\text{text}}$.

We show a typical example of a standard text-only instruction and interleaved instruction:

```
Text-only:  <obs> Place [the blue spoon near microwave] into [silver pot
on towel].
Interleaved image-text:  <obs> Place [image1      ] into [image2    ].
```

where <obs> stands for observation image(s), and [image1    ] and [image2    ] are interleaved instruction images. As shown in Figure 1, Interleave-VLA supports more flexible formats.

## 3.2 INTERLEAVE-VLA PARADIGM

Outlined in Figure 2, our Interleave-VLA paradigm, designed for generating continuous actions in the real world from interleaved inputs, comprises three core components: **(1)** a straightforward and effective adaptation module, **(2)** a scalable training process tailored for interleaved data, and **(3)** a versatile inference interface that supports interleaved instructions.

**Adaptation.** Interleave-VLA extends Text-VLA by introducing special separator tokens into the tokenizer of the base VLA model, allowing it to distinguish between image and text tokens. The input processor is updated to support the interleaved format, while the core VLA architecture re-

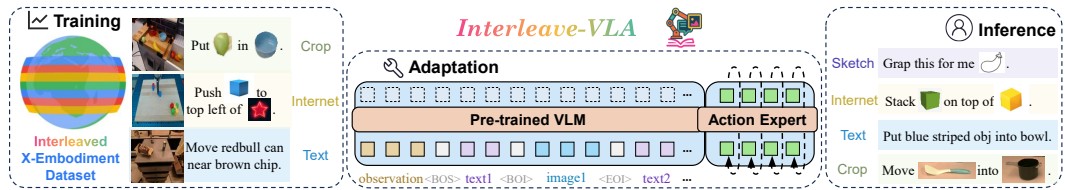

Figure 2: Overview of the Interleave-VLA paradigm, featuring an extendable adaptation of Text-VLA to handle interleaved inputs, scalable training on a constructed large interleaved dataset, and versatile inference that supports a wide range of interleaved instructions.

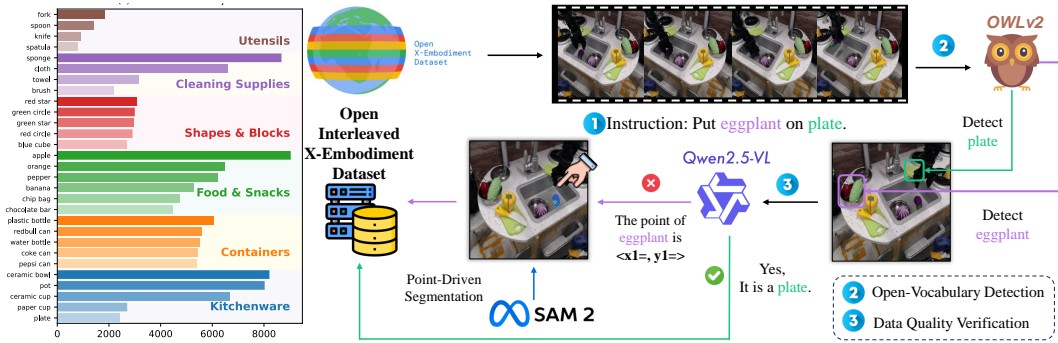

Figure 3: **Left:** Our open interleaved X-Embodiment dataset features a large number of high-quality cropped images with diversity across objects. **Right:** Interleave dataset generation pipeline: (1) Instruction Parsing: use LLM to extract key objects from language instructions. (2) Open-Vocabulary Detection: use OWLv2 to locate and crop target objects from trajectory frames based on the parsed instruction keywords. (3) Data Quality Verification: use Qwen2.5-VL to verify the detected objects, and if needed, provide keypoints for more precise segmentation using Segment Anything.

mains unchanged. This paper focuses on applying Interleave-VLA to $\pi_0$ (Black et al., 2024), a state-of-the-art Text-VLA model. Despite its pretrained Paligemma (Beyer et al., 2024) lacking native support for interleaved data, Interleave-VLA enables this capability. Our adaptation is effective in enhancing the zero-shot generalization potential of $\pi_0$. Moreover, the simplicity of this adaptator makes it easily applicable to other VLA models, such as OpenVLA (Kim et al., 2024). For further details on our model-agnostic adaptation, see Appendix D.

**Training.** We train the interleaved-adapted $\pi_0$ model using the large-scale interleaved embodied dataset from Section 3.3, without modifying its hyperparameters or flow matching objective. The training process efficiently scales with the dataset size and cross-modal instruction diversity.

**Inference.** Interleave-VLA paradigm supports both text and interleaved instructions during inference, with interleaved inputs significantly improving generalization to unseen scenarios. Interleaved images in prompts offer great versatility, as they can be sourced from diverse sources such as robot camera crops, web images, or hand-drawn sketches, even when the image styles differ from those in the robot's training data. To simplify interaction with the robot, we also design a user-friendly GUI.

### 3.3 CONSTRUCTION OF OPEN INTERLEAVED X-EMBODIMENT DATASET

A large-scale and high-quality pretraining dataset scales up vision-language-action (VLA) (Black et al., 2024; Brohan et al., 2023; Kim et al., 2024) training. However, current real-world datasets only include text instructions and thus do not support Interleave-VLA training. We consequently design a unified pipeline to automatically relabel and generate interleaved data across diverse datasets.

Our overall dataset generation pipeline consists of three main steps: instruction parsing, open-vocabulary detection, and data quality verification, as illustrated in Figure 3. **First**, we use Qwen2.5 (Yang et al., 2024) to extract key objects from language instructions. Unlike rule-based NLP tools like SPaCy (Honnibal, 2017), Qwen2.5 adapts to diverse instruction formats without requiring case-specific rules. It also effectively summarizes lengthy instructions, such as those in datasets like (Shah et al., 2023). **Second**, for open-vocabulary detection, we use the open-vocabulary

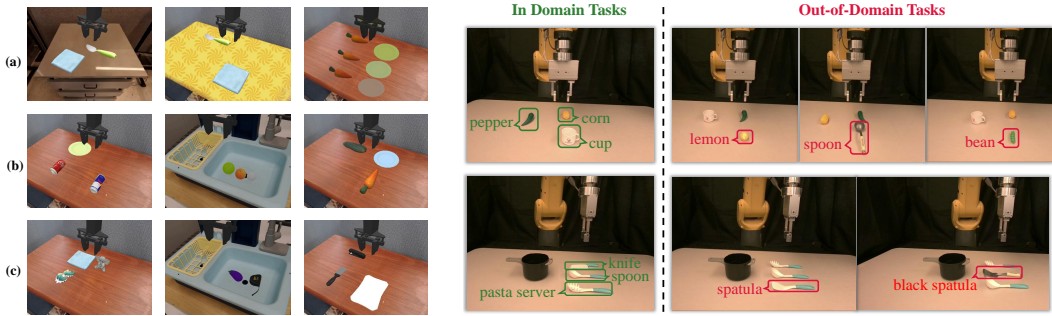

Figure 4: **Left**: Illustration of generalization settings in SIMPLER. (a) Visual generalization: unseen environments, tablecloths, and lighting conditions. (b) Semantic generalization with novel objects from known categories. (c) Semantic generalization with objects from entirely new categories not seen during training. **Right**: Real-world generalization experiments. In-Domain and out-of-Domain settings in the real world on a FANUC LRMate 200iD/7L robotic arm.

detector OWLv2 (Minderer et al., 2023) to locate and crop target objects from trajectory frames based on the parsed instruction keywords, achieving 82.6% accuracy. **Finally**, we introduce data quality verification for harder cases where OWLv2 fails: Qwen2.5-VL (Bai et al., 2025) verifies the detected objects, and if needed, provides keypoints for more precise segmentation using Segment Anything (Ravi et al., 2024). This collaborative approach leverages the complementary strengths of VLMs, raising accuracy to 95.6%. Detailed metrics and analysis are provided in Appendix E.

We release a large-scale interleaved cross-embodiment dataset in the real world, featuring diverse tasks and instructions. This dataset integrates 11 datasets from Open X-Embodiment (O'Neill et al., 2024), including RT-1 (Brohan et al., 2022), Berkeley Autolab UR5 (Chen et al.), IAMLab CMU Pickup Insert (Saxena et al., 2023), Stanford Hydra (Belkhale et al., 2023), UTAustin Sirius (Liu et al., 2023b), Bridge (Walke et al., 2023a), Jaco Play (Dass et al., 2023), UCSD Kitchen (Yan et al., 2023), BC-Z (Jang et al., 2022), Language Table (Lynch et al., 2023), and UTAustin Mutex (Shah et al., 2023). The curated dataset comprises 210k episodes and 13 million frames, spanning 3,500 unique objects and a wide range of task types. To enhance instruction diversity, we augment our dataset by randomly integrating Internet-sourced images alongside the original text instructions.

# 4 EXPERIMENTS

In the experiments, our aim is to answer the following research questions:

(1) How does our Interleave-VLA paradigm compare to conventional Text-VLA paradigm?

(2) What are the common failure modes of Text-VLA, and how does Interleave-VLA address them?

(3) What are the benefits of the different design choices in our Interleave-VLA paradigm?

## 4.1 SIMULATION COMPARISON WITH TEXT-VLA

**Task setup.** We use SimplerEnv (Li et al., 2024), a real-to-sim evaluation suite, to efficiently evaluate policies in realistic scenarios. Performances are tested on SimplerEnv-Bridge setup, which uses a WidowX robot configuration compatible with the BridgeData V2 (Walke et al., 2023b). Since SimplerEnv is built for Text-VLA, to enable scalable evaluation of Interleave-VLA models, we extend SimplerEnv with interleaved image–text prompts via automated pipeline (Section 3.3).

In addition to the four SimplerEnv-Bridge tasks from BridgeData V2, we include ten new tasks for generalization evaluation. Following commonly used Stone et al. (2023), these tasks focus on assessing both *visual generalization* and *semantic generalization*. *Visual generalization* assesses robustness to novel foreground, lighting, and environment backgrounds. *Semantic generalization* assesses the model's ability to manipulate in novel scenarios and the presence of diverse distractors. This evaluation is further divided into two sub-categories: **(1)** novel objects from previously seen categories, and **(2)** objects from entirely unseen categories. See left part of Figure 4 for an overview.

Table 2: **Interleave-VLA and Text-VLA comparison on SimplerEnv**. **In-Domain** includes 4 tasks following SimplerEnv-Bridge setup. We add 3 **Out-of-Domain** evaluation suites, namely: Visual, Novel Object, and Novel Category. $\pi_0$ with full adaptation (Interleave-VLA Full) performs better than $\pi_0$ with no adaptation (Text-VLA) by over $2\times$ in out-of-domain semantic generalization tasks. It also outperforms $\pi_{0.5}$ which enjoys additional pretraining with additonal object grounding and detection VQA data. Results are evaluated on 3 seeds. We use **bold** and underline to represent the $1^{st}$ and $2^{nd}$ highest. For quantitative breakdown of failure modes, please refer to Figure 11.

| Base Model | Paradigm | Train/Eval Modality | In-Domain | Out-of-Domain | | | |
|---|---|---|---|---|---|---|---|
| | | | | Visual | Novel Object | Novel Category | Avg. |
| RT-1-X (O'Neill et al., 2024) | Text-VLA | Text/Text | $1.1 \pm 0.5$ | $0.0 \pm 0.0$ | $3.5 \pm 0.4$ | $5.8 \pm 0.3$ | $3.2 \pm 0.2$ |
| Octo (Team et al., 2024) | Text-VLA | Text/Text | $17.4 \pm 1.3$ | $12.5 \pm 0.1$ | $10.8 \pm 0.7$ | $8.2 \pm 0.2$ | $10.5 \pm 0.3$ |
| Spatial-VLA (Qu et al., 2025) | Text-VLA | Text/Text | $38.4 \pm 1.5$ | $19.6 \pm 0.0$ | $17.1 \pm 0.0$ | $17.6 \pm 0.0$ | $18.0 \pm 0.0$ |
| $\pi_{0.5}$ (Intelligence et al., 2025) | Text-VLA | Text/Text | $57.2 \pm 3.9$ | $53.9 \pm 1.1$ | $\underline{50.9 \pm 0.3}$ | $\underline{41.8 \pm 0.5}$ | $\underline{49.0 \pm 0.5}$ |
| $\pi_0$ (Black et al., 2024) | Text-VLA | Text/Text | $68.1 \pm 1.3$ | $72.4 \pm 1.1$ | $26.0 \pm 3.6$ | $19.3 \pm 1.5$ | $39.5 \pm 0.5$ |
| $\pi_0$ (Black et al., 2024) | Interleave-VLA (Partial) | Interleave/Text | $\underline{70.1 \pm 0.9}$ | $\mathbf{76.8 \pm 0.2}$ | $35.8 \pm 0.2$ | $20.9 \pm 1.9$ | $43.6 \pm 0.6$ |
| $\pi_0$ (Black et al., 2024) | Interleave-VLA (Full) | Interleave/Interleave | $\mathbf{70.5 \pm 1.3}$ | $\underline{73.2 \pm 0.3}$ | $\mathbf{53.8 \pm 1.5}$ | $\mathbf{57.3 \pm 2.8}$ | $\mathbf{60.6 \pm 1.1}$ |

**Baselines.** We evaluate Interleave-VLA against leading Text-VLA models, including RT-1-X (Brohan et al., 2022), Octo (Team et al., 2024), and SoTA $\pi_0$ (Black et al., 2024). To directly compare with Text-VLA, we implement Interleave-VLA (Partial) on $\pi_0$, which is trained with interleaved inputs but tested with text-only instructions. Finally, we evaluate the full potential of the Interleave-VLA paradigm, where $\pi_0$ is both trained and tested using interleaved inputs. Both Interleave-VLA and Text-VLA paradigms are trained on trajectories from BridgeData V2 (Walke et al., 2023a).

**Results.** In-domain results in Table 2 show that our Interleave-VLA paradigm performs on par with Text-VLA for familiar tasks, demonstrating that interleaved instructions are interpretable thanks to Interleave-VLA training process. For out-of-domain tasks, Interleave-VLA (Partial) already outperforms Text-VLA, benefiting from the multimodal nature of interleaved data, which helps mitigate overfitting. The full Interleave-VLA further enhances generalization, achieving $2\times$ better performance on semantically out-of-domain tasks.

**Analysis.** The substantial performance gains of Interleave-VLA (Full) over Text-VLA and Interleave-VLA (Partial) mainly stem from the explicit visual grounding supplied by interleaved instruction images, which reduces a phenomenon we term **attentional hallucination**. To qualitatively illustrate this, we compute the attention scores of target object tokens relative to the tokenized observation in out-of-domain settings. As shown in Figure 5, we identify three primary failure patterns: **(1)** *Attentional Bias*, where focus is incorrectly allocated to prominent distractor instead of the target object; **(2)** *Diffused Attention*, characterized by a complete lack of a focal point as attention spreads thinly across the entire scene, suggesting model uncertainty; and **(3)** *Attention Leakage*, where the model correctly identifies the target but its focus is not tightly contained, scattering onto irrelevant background areas. These failures can be attributed to semantic ambiguity in cluttered visual contexts and distributional bias in the training data. For example, **semantic ambiguity** arises when the instruction says "toy dinosaur" but a similarly shaped toy elephant is present, the text-only Text-VLA model often makes an arbitrary choice; **distributional bias** manifests when Text-VLA misidentifies a Coca-Cola can as a Red Bull can because the rare token "redbull" is segmented into "red" + "bull", causing it to over-attend to "red" and biasing attention toward the familiar red Coca-Cola can. These built-in biases are difficult to address with conventional Text-VLA. In contrast, Interleave-VLA outperforms Text-VLA baselines *by leveraging in-context visual grounding and cross-modality training to reduce attentional hallucinations*.

## 4.2 REAL ROBOT COMPARISON

**Task setup.** We evaluate on a FANUC LRMate 200iD/7L robotic arm equipped with an SMC gripper. Two manipulation tasks are considered: (1) picking up food or fruits, and (2) picking and placing kitchenware. To assess *semantic generalization*, we follow the SimplerEnv setup. See the right part of Figure 4 for an overview, with additional details in Appendix G.3.

**Baselines.** All baselines use the same base VLA model $\pi_0$, with two following the Interleave-VLA paradigm and one using Text-VLA. Each baseline is finetuned on 20 teleoperated demonstrations per object, collected using a space mouse. Optionally, pretraining is performed on the robot data in Section 3.3, training Text-VLA on text instructions and Interleave-VLA on interleaved instructions.

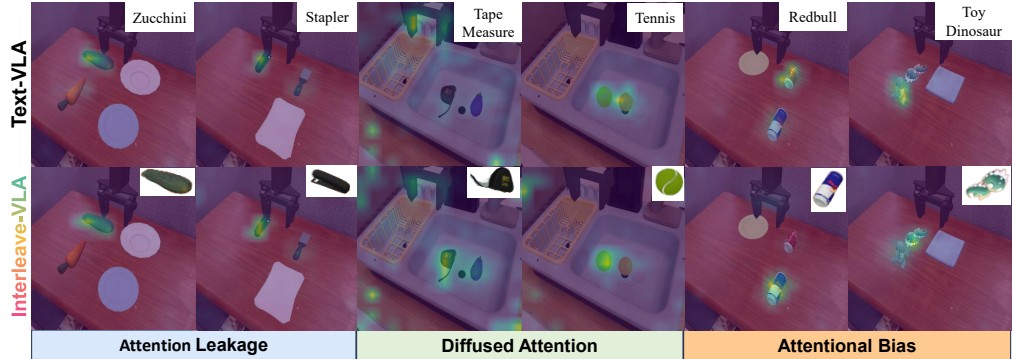

Figure 5: **Qualitative analysis of Interleave-VLA's improved performance over the Text-VLA paradigm.** In out-of-domain SimplerEnv tasks with unfamiliar objects, Text-VLA displays **attentional hallucination**, which typically manifests in three categories: **(1)** *Attention Leakage*: the target is partially attended, but focus spills onto irrelevant background or distractor regions; **(2)** *Diffused Attention*: attention is broadly scattered with no dominant focus, indicating uncertainty about the target; **(3)** *Attentional Bias*: attention centers on a salient distractor instead of the true target. Interleave-VLA effectively mitigates these issues by leveraging in-context visual cues from interleaved instructions, demonstrating consistent attention on target objects.

Table 3: Comparison of success rates (Succ) and correct object picking rates (Acc) in real-robot experiments. All the baselines use the base VLA model $\pi_0$. Interleave-VLA adapted achieves **2-3× higher out-of-domain performance** compared to Text-VLA. "PT" indicates pretraining on our interleaved dataset built in Section 3.3. Notably, although the pretraining dataset does not include FANUC robot arm data, it still enables strong cross-embodiment transfer to FANUC.

| Paradigm | PT | In-Domain | | | | | | | | Out-of-Domain | | | | | |
| | | pepper | | corn | | cup | | Avg | | bean | | lemon | | Avg | |
| | | *Succ.* | *Acc.* | *Succ.* | *Acc.* | *Succ.* | *Acc.* | *Succ.* | *Acc.* | *Succ.* | *Acc.* | *Succ.* | *Acc.* | *Succ.* | *Acc.* |
| Interleave-VLA | ✗ | 17 | 33 | 0 | 33 | 0 | 33 | 6 | 33 | 0 | 40 | 0 | 33 | 0 | 37 |
| Text-VLA | ✓ | 58 | 83 | 33 | 100 | 25 | 100 | 39 | 94 | 8 | 8 | 17 | 42 | 13 | 25 |
| Interleave-VLA | ✓ | 58 | 100 | 75 | 100 | 67 | 100 | 67 | 100 | 75 | 100 | 67 | 100 | 71 | 100 |
| | | pasta server | | spoon | | knife | | Avg | | spatula | | black spatula | | Avg | |
| | | *Succ.* | *Acc.* | *Succ.* | *Acc.* | *Succ.* | *Acc.* | *Succ.* | *Acc.* | *Succ.* | *Acc.* | *Succ.* | *Acc.* | *Succ.* | *Acc.* |
| Interleave-VLA | ✗ | 33 | 67 | 8 | 58 | 17 | 58 | 19 | 61 | 0 | 67 | 0 | 50 | 0 | 59 |
| Text-VLA | ✓ | 58 | 83 | 58 | 75 | 33 | 58 | 50 | 72 | 8 | 8 | 33 | 42 | 21 | 25 |
| Interleave-VLA | ✓ | 50 | 67 | 58 | 83 | 33 | 58 | 47 | 69 | 25 | 100 | 50 | 67 | 38 | 84 |

**Results.** Table 3 shows that Interleave-VLA achieves **2-3× higher out-of-domain performance** compared to Text-VLA when both are pretrained. Unlike the SimplerEnv experiments, where large-scale BridgeData V2 supports strong performance, the real-robot setup relies on a smaller self-collected dataset. In this low-data regime, directly finetuning $\pi_0$ performs poorly (first row). Pretraining on the Interleaved X-Embodiment dataset significantly boosts performance through effective cross-embodiment transfer, reducing the need for laborious data collection.

### 4.3 ANALYSIS OF INTERLEAVE-VLA'S GENERALIZATION AND EMERGENT CAPABILITIES

#### 4.3.1 INTERLEAVE-VLA ADAPTATION: EXTENDING TO OTHER VLA MODELS

The model-agnostic design of Interleave-VLA allows easy adaptation to other VLA models, demonstrating its effectiveness in enhancing manipulation generalization across diverse architectures. To validate this, we extend Interleave-VLA to OpenVLA (Kim et al., 2024), a state-of-the-art VLA model with a distinct architecture and training objective compared to $\pi_0$. We evaluate it on VIMA-Bench (Jiang et al., 2023), which includes four levels of manipulation planning tasks involving objects with irregular, cartoon-like shapes and textures. As shown in Figure 6, we compare Interleave-VLA against several end-to-end baselines, including the Text-VLA model OpenVLA and other VIMA-like models such as VIMA-Gato, VIMA-Flamingo, and VIMA-GPT (Jiang et al., 2023). Across all four generalization levels, our general Interleave-VLA paradigm, when directly extended to OpenVLA, achieves the best performance without relying on any task-specific designs. Details on the adaptation and evaluation are provided in Appendices D.2 and G.2.

Table 4: Interleave-VLA unlocks powerful **zero-shot** generalization to diverse instruction modalities, including hand-drawn sketches, user-cropped images, and Internet photos, **without ever seeing them in training dataset**. The consistently high accuracy demonstrates that Interleave-VLA can robustly interpret and execute visually grounded instructions, showing strong potential for flexible and practical human-robot interaction. For more styles of sketches and potential failure modes, please refer to Table 8 and 9 in Apppendix C.

| Task | Prompt A | A Succ. (%) | A Acc. (%) | Prompt B | B Succ. (%) | B Acc. (%) |
|---|---|---|---|---|---|---|
| | | 58.3 | 90.0 | | 48.8 | 86.0 |
| | | 75.8 | 100 | | 58.8 | 100 |
| | | 71.7 | 100 | | 80.8 | 100 |
| | | 70.0 | 96.0 | | 73.3 | 100 |
| | | 69.6 | 100 | | 76.3 | 100 |
| | | 75.5 | 100 | | 71.7 | 100 |

### 4.3.2 INTERLEAVE-VLA INFERENCE: FLEXIBILITY AND EMERGENT GENERALIZATION

The inference interface of Interleave-VLA, which we show effectively reduces attentional hallucination problem, is highly versatile. Interleave-VLA demonstrates strong performance across diverse ways of specifying instructions in VIMA-Bench, including goal image matching and multi-image instruction following. These results demonstrate the flexibility of the Interleave-VLA paradigm, driven by its unified image-text interleaved instructions for general robotic manipulation.

Building on its versatile inference interface, Interleave-VLA further showcases an emergent capability to interpret instructions in a completely **zero-shot manner**, directly handling unseen input modalities without any additional finetuning. Table 4 demonstrates the examples of image instruction types and their corresponding high performance. Instructions can be in diverse formats, including: (1) **Cropped Image Instructions:** Users can directly crop a region from the screen to indicate the target object. (2) **Internet Image Instructions:** Users may supply any image—such as a photo retrieved from the Internet—to represent the desired object. (3) **Hand-Drawn Sketch Instructions:** Users can draw sketches or cartoons about their intentions.

The interleaved instruction format naturally accommodates diverse input types, making human-robot interaction more intuitive by removing the need for users to precisely describe complex objectives with detailed text. This flexibility significantly enhances the model's ability to generalize, as evidenced by the improvements observed in both in-domain and out-of-domain tasks, where interleaved image-text instructions effectively reduce attentional hallucinations in VLA models. These advancements in Interleave-VLA collectively pave the way for more adaptable robotic systems.

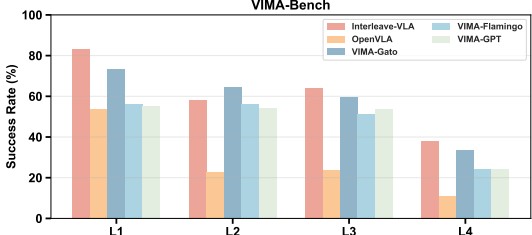

Figure 6: **VIMA-Bench** results across four generalization levels: L1 (object placement), L2 (novel combination), L3 (novel object), and L4 (novel task). To demonstrate the extendability of our Interleave-VLA paradigm, we apply it to another Text-VLA model, OpenVLA. Interleave-VLA outperforms Text-VLA by $2\times$ across all difficulty levels, further highlighting its superior generalization capabilities with evidence from new task sets and a different base VLA model.

### 4.3.3 INTERLEAVE-VLA TRAINING: MODALITY AND INSTRUCTION DIVERSITY MATTER

The most obvious scaling law of Interleave-VLA is dataset size, which is shown in large data domain (Appendix H) and low-data domain (Table 3). Overall, our results underscore the importance of the curated large-scale Interleaved X-Embodiment Dataset (Section 3.3) in fostering robust and generalizable Interleave-VLA. In this section, we delve deeper into the training data and identify two key factors that drive scalability and generalization: **(1)** the modality diversity of the dataset and **(2)** the diversity of prompt images.

The **diversity of modalities** in training dataset is crucial for achieving robust performance VLAs, particularly for out-of-domain generalization. This principle is empirically demonstrated by com-

Table 5: **Importance of prompt image diversity for Interleave-VLA.** "In-Domain" stands for seen tasks; "Out-of-Domain" reports unseen scenarios. Combining both task-specific and Internet images as prompts achieves the best overall performance.

| Data | In-Domain | Out-of-Domain |
|------|-----------|---------------|
| Internet Only | 59.2 | 69.1 |
| Task-specific Only | 67.5 | 67.1 |
| Mixed | **71.0** | **71.7** |

Table 6: **Interleaved instructions contribute through both format and content.** Visual-goal cues drive out-of-domain generalization by providing explicit image information, while the interleaved format offers complementary gains and prevents task ambiguity, especially for underspecified goals such as "Move Near". See Section 4.4 for details.

| Instruction (train and inference) | In-Domain | Unseen BG | Unseen Obj | Unseen Cat. | Move Near |
|-----------------------------------|-----------|-----------|------------|-------------|-----------|
| Text | 69.2 | 71.4 | 30.2 | 21.0 | 66.6 |
| Visual Goal | 67.8 | **74.6** | 48.0 | 51.9 | 0.0 |
| Interleaved Img-Text | **71.3** | 73.4 | **53.9** | **54.2** | **68.8** |

paring the performance of Interleave-VLA (Partial) and Text-VLA, which share an identical architecture (see Table 2). $\pi_0$ trained with cross-modal, interleaved image-text instructions achieves absolute improvements of $+2.5$ on familiar in-domain tasks and a more substantial $+5.7$ on tasks requiring generalization to new objects. We attribute this performance gain to the development of richer multimodal representations by mitigating the model's tendency to overfit to unimodal text signals (Alayrac et al., 2022).

**Instruction image diversity** is crucial as well, Table 5 demonstrates that combining Internet images with task-specific images cropped from robot observations yields the best overall performance. Using only Internet images leads to lower in-domain accuracy due to limited task relevance, while relying solely on cropped images improves in-domain results but lacks diversity. Mixing both sources provides complementary advantages, resulting in enhanced accuracy and stronger generalization.

## 4.4 INTERLEAVED INSTRUCTION: BOTH FORMAT AND CONTENT ARE CRUCIAL

The interleaved image–text instruction in Interleave-VLA contains two separable factors: (1) the format, which allows image and text tokens to appear in arbitrary order, and (2) the content, which introduces explicit visual goal cues. Both contribute to generalization, but in different ways.

To isolate the contribution of the visual goal signal, we perform an ablation in the SimplerEnv-Bridge setting (Table 2). In Table 6, We compare our interleaved image–text content ("Put [Object A] to [Object B]") with a pure visual-goal content ("[Object A][Object B]"). The first four columns confirm that adding visual goal cues improves performance. However, the interleaved format itself is equally crucial, as highlighted by the "Move Near" column. Under the Visual Goal format, the model consistently misinterprets the task as a "put on" operation. This occurs because many common robot-instruction templates, e.g., "Put object A [near / to the left of / on] object B", become ambiguous when expressed only through object images. Since "put on" primitives are far more prevalent than "move near" in the training data, the Visual Goal format tends to collapse to this dominant interpretation. In contrast, the interleaved format provides an unambiguous representation that is necessary for leveraging embodied data across diverse manipulation tasks.

In summary, interleaved content offers image-text grounding, whereas the interleaved format enables task unification with minimal ambiguity.

## 5 CONCLUSION

Text-only instructions in most robotic policies can be insufficient for unseen scenarios and even causes hallucinations. To address this, we propose Interleave-VLA, a simple and effective paradigm for adapting existing Text-VLA models to process interleaved image-text instructions. To overcome the lack of real-world interleaved datasets, we develop an automatic pipeline that generates a large-scale dataset with 210k episodes and 13 million frames from Open X-Embodiment. With minimal modifications to current VLA models, Interleave-VLA achieves $2\times$ improvement in generalization across both simulation and real-world experiments. Furthermore, our approach demonstrates strong emergent zero-shot generalization to diverse user instructions never seen during training—including hand-drawn sketches, cropped images, and Internet photos—making it both practical and flexible for real-world robotic applications.

**Limitations.** Please refer to Appendix B for detailed discussion.

ETHICS STATEMENT

This research adheres to the ICLR ethical guidelines and upholds the principles of responsible research, We ensure that no personally identifiable, sensitive, or harmful data were used. Our experiments were based on publicly available datasets and did not involve any human subjects or vulnerable groups. We have considered the potential societal impact of our methods, including the risk of misuse, and believe that these contributions primarily advance scientific understanding and do not pose foreseeable harm.

REPRODUCIBILITY STATEMENT

We follow the reproducibility guidelines in the ICLR 2026 author guidelines. We will open source code, configuration files, and scripts to reproduce our results, including dataset construction, model training, and evaluation, on platforms such as GitHub and Huggingface as soon as possible.

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

# Appendix

## A    THE USE OF LARGE LANGUAGE MODELS (LLMS)

We affirm that this paper is prepared and written entirely by us. We did not use any large language models (LLMs) to generate the abstract, content, or any substantive part of the text. All ideas, analysis, and conclusions are the sole product of the authors' original thought and research. LLMs were employed solely for polishing grammar and refining phrasing, similar in scope to conventional grammar or style checkers.

## B    LIMITATIONS OF INTERLEAVE-VLA

While Interleave-VLA exhibits strong generalization, training and deployment with interleaved inputs increases computational cost due to longer image-token sequences. Nevertheless, as discussed in Appendix D.3, this added cost is minimal in most practical scenarios. Regarding further improvements, future work could investigate more efficient image-token compression strategies and extend VLA models to support interleaved outputs—such as text or images—alongside actions, a direction recent studies suggest may further improve performance (Intelligence et al., 2025; Zhao et al., 2025a). Moreover, from a data-pipeline perspective, although not readily observable in this paper, certain failure modes may potentially degrade Interleave-VLA performance in edge cases. Please refer to Figure 7 and Table 10 for a detailed failure case analysis.

## C    INTERLEAVE-VLA TRULY UNDERSTANDS OR VISUALLY OVERFITS?

It is of value to know whether Interleave-VLA truly grounds its decisions in visual inputs, or whether it overfits to certain spurious correlations in the image or the prompt format itself. To this end, we design controlled probes that explicitly disentangle textual and visual information and analyze how Interleave-VLA behaves under alignment and contradiction. It is important to understand whether Interleave-VLA truly grounds its decisions in visual inputs, or instead overfits to spurious correlations in the image or in the prompt format itself. To this end, we design controlled probes that explicitly disentangle textual and visual information and analyze how Interleave-VLA behaves under both alignment and contradiction between the two modalities.

**Understanding image–text contradiction.** To study grounding under conflicting modalities, we consider a color-conditioned pick-and-place task: "Pick up a [green/yellow] block and place it on the towel." We compare five instruction formats (illustrated using the "green block" case):

1. **text**: "Pick up the green block . . . "
2. **interleave**: "Pick up the `[Image of green block]`..."
3. **interleave-aligned**: "Pick up the green block `[Image of green block]`..."
4. **interleave-contradict (image correct, text wrong)**: "Pick up the yellow block `[Image of green block]`..."
5. **interleave-contradict (image wrong, text correct)**: "Pick up the green block `[Image of yellow block]`..."

We report two metrics: (1) *Success Rate*, the fraction of episodes in which the correct block is picked and placed; and (2) *Intention Accuracy*, the fraction of episodes in which the chosen block matches the task objective. Results are summarized in Table 7.

From these comparisons, we observe that Interleave-VLA is able to jointly interpret image and text modalities (first three columns). In the last two columns, it is notable that Interleave-VLA consistently prioritizes textual instructions over visual cues when the two conflict. This behavior is clearly reflected in the *interleave-contradict-v2* setting (image wrong, text correct), where the model achieves 100% intention accuracy, and in the *interleave-contradict-v1* setting (image correct, text wrong), where the intention accuracy drops to 0%. (The small 4.2% execution success rate in this setting stems from cases where the model initially places the wrong object on the towel but

| Metrics / Format | Text | Interleave | Interleave-Aligned | Interleave-Contradict (image correct, text wrong) | Interleave-Contradict (image wrong, text correct) |
|---|---|---|---|---|---|
| Success Rate (yellow / green) | 93.8% / 97.9% | 95.8% / 93.8% | 89.6% / 93.8% | 4.2% / 0.0% | 62.5% / 89.6% |
| Intention Accuracy (yellow / green) | 100.0% / 100.0% | 100.0% / 100.0% | 100.0% / 100.0% | 0.0% / 0.0% | 100.0% / 100.0% |

Table 7: **Grounding under image–text contradiction.** Success Rate and Intention Accuracy (in %) of Interleave-VLA on the color-conditioned block manipulation task under different instruction formats. First 3 columns show that Interleave-VLA understands multimodality accurately and last 2 columns imply that Interleave-VLA consistently learns to attend to text when modalities contradict each other, which is evidently not driven by hallucination.

| Metrics / Style | Normal | OCR | Quick | Abstract | Misleading | Ambiguous |
|---|---|---|---|---|---|---|
| **Success Rate** (yellow / green) | 95.8% / 89.6% | 93.8% / 91.7% | 91.7% / 81.3% | 56.3% / 14.6% | 37.5% / 8.3% | 20.8% / 16.7% |
| **Intention Accuracy** (yellow / green) | 100.0% / 100.0% | 100.0% / 100.0% | 100.0% / 91.7% | 70.8% / 20.8% | 56.3% / 8.3% | 35.4% / 66.7% |

Table 8: **Performance across sketch styles.** Success Rate and Intention Accuracy (in %) of Interleave-VLA when the target object is specified by sketches with different levels of clarity and ambiguity.

subsequently corrects its action.) Overall, these results indicate that the model genuinely understands the combined image–text prompt rather than merely overfitting to specific visual patterns in the prompt image, as further supported by the new Tables 9 and 10 in the updated manuscript.

**Understanding complex sketches.** To further probe visual grounding beyond simple object photos, we revisit the task in Q1 and replace the reference image with human-created sketches of varying informativeness and ambiguity. We categorize sketches into the following styles:

1. **Normal**: Detailed sketches drawn by humans in approximately 15 seconds.

2. **OCR**: Simple drawings of a square/cube with the caption "yellow" or "green".

3. **Quick**: Rough sketches produced in under 5 seconds.

4. **Abstract**: Minimalistic sketches of a square/cube labeled only with "G" or "Y" (for green/yellow).

5. **Misleading**: Intentionally confusing sketches, e.g., a yellow-outlined cube with the caption "green".

6. **Ambiguous**: Under-specified sketches that omit clear color information or only loosely highlight the target object (e.g., a rough circle around a region in the scene).

Table 8 reports *Success Rate* and *Intention Accuracy* for each style. As sketches become less informative and require more common-sense reasoning to interpret, performance degrades substantially.

These results show that Interleave-VLA can reliably interpret rich, well-formed sketches (Normal, OCR, Quick), but its performance drops with highly abstract, misleading, or ambiguous sketches, where successful grounding requires additional high-level reasoning about the sketch creator's intent.

**Conclusion.** Taken together, the contradiction and sketch experiments provide strong evidence that Interleave-VLA *truly understands* visual prompts, including shape and color (Normal), content (OCR sketches), and cross-modal consistency (Interleave-Contradict). The model is not simply treating the image as a generic conditioning token tied to the accompanying text: in our experiments, Interleave-VLA successfully executes instructions when the target object is specified *either* in text *or* in the image, including cases where the two modalities disagree. At the same time, the degradation on abstract and ambiguous sketches highlights an important avenue for future work: equipping interleaved vision–language–action models with stronger reasoning capabilities to robustly interpret under-specified or systematically misleading visual prompts.

# D    INTERLEAVE-VLA IMPLEMENTATION DETAILS

We extend two state-of-the-art VLA models, $\pi_0$ (Black et al., 2024) and OpenVLA (Kim et al., 2024), to develop Interleave-VLA. While VLA models encompass a wide range of architectures (Intelligence et al., 2025; Team et al., 2025; Bjorck et al., 2025; Liu et al., 2024b; Shi et al., 2024;

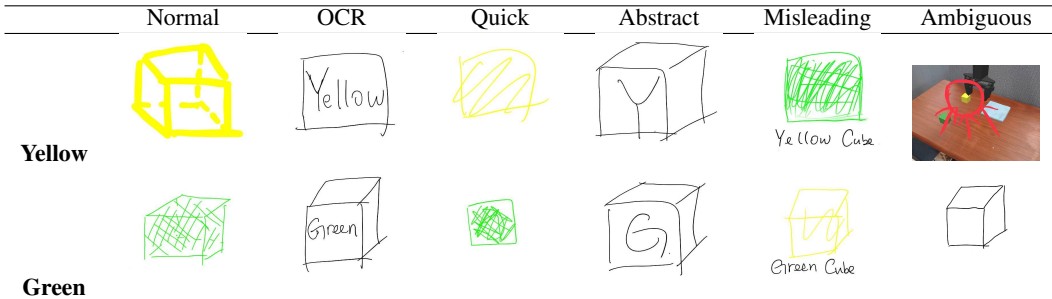

|  | Normal | OCR | Quick | Abstract | Misleading | Ambiguous |
|---|---|---|---|---|---|---|
| **Yellow** | | | | | | |
| **Green** | | | | | | |

Table 9: Example sketches for each style. Yellow-cube and green-cube sketches representing each drawing style. The examples are sourced from multiple individuals and are designed to probe Interleave-VLA's ability to understand different aspects of a single object.

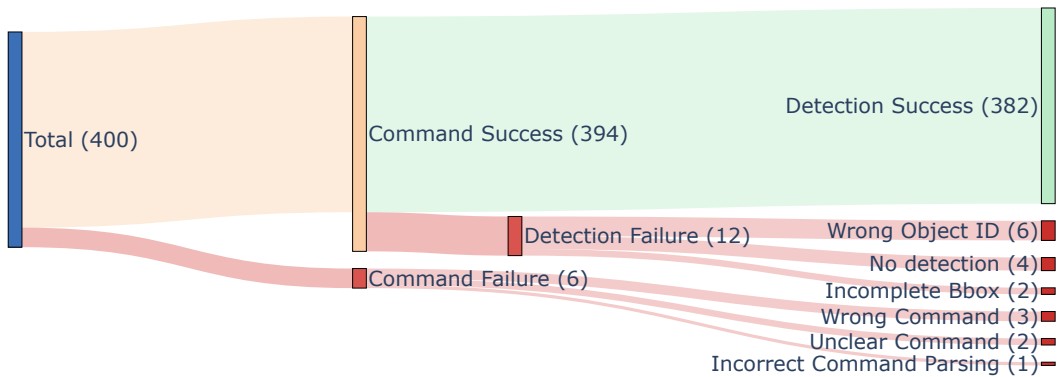

Figure 7: **Interleave-VLA data-pipeline failure analysis.** We expand the evaluation to 400 examples and quantify failures across the pipeline stages, from command parsing to visual grounding. The majority of samples (394/400) pass the command-parsing stage, while detection failures account for most of the remaining errors (12/18). Each branch is annotated with the dominant failure modes, including wrong object identity, missed detections, incomplete bounding boxes, ambiguous commands, and incorrect command parsing. For example driven reviews, please refer to Table 10.

Brohan et al., 2022; 2023; Team et al., 2024; Chi et al., 2023), we focus on those based on VLM backbones due to their inherent ability to process image-text pairs. However, our approach is not restricted to VLM-based methods and can be extended to other sequence modeling approaches for action prediction (Chi et al., 2023; Team et al., 2024; Liu et al., 2024b; Brohan et al., 2022). The key modification involves interleaving image and text embeddings within the input sequence. Investigating the feasibility of this modification for other sequence modeling VLAs is an exciting direction for future research. In this work, we focus on and provide adaptations of Interleave-VLA from $\pi_0$ and OpenVLA in the following sections in more detail.

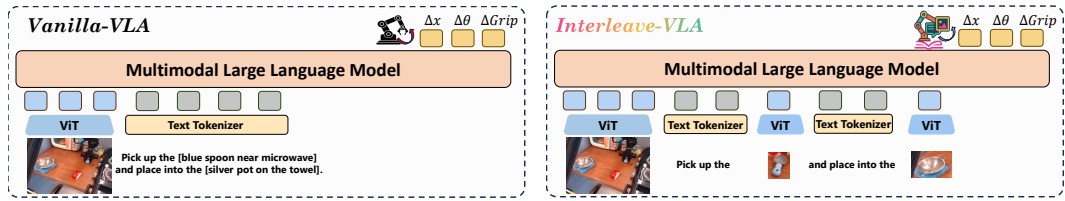

Figure 8: Comparison of Interleave-VLA and Text-VLA architectures. Interleave-VLA is model-agnostic and requires minimal modifications to existing VLA architectures. The only change is the input format, which allows for interleaved image-text instructions.

| Failure Mode | Original instruction | Task image | Expected result |
|---|---|---|---|
| **No Detection** | `move light switch right` |  |  |
| **Wrong Object ID** | `take cucumber out of cup` |  |  |
| **Incomplete BBox** | `move the yellow block from the top of the stack to the table` |  |  |
| **Wrong Command** | `place the carrot in the middle` |  | Carrot is not present in the scene (only sushi). This comes from errors of original Open-X dataset. |
| **Unclear Command** | `put the object in the drawer` |  | The instruction is ambiguous: it does not specify which object to put into the drawer. A disambiguated instruction (e.g., "put the *red block* in the drawer") is expected. This comes from errors of original Open-X dataset. |
| **Incorrect Command Parsing** | `move the blue curl block from the cup to the top of the cube` |  | The LLM parser fails to treat "blue curl block" as a single object. Therefore, the object detection module is unable to retrieve the correct blue block from other colors. |

Table 10: **Qualitative examples of primary failure modes in the dataset-construction pipeline.** For each error type, we show the original natural-language instruction, the corresponding task image, and the analysis of expected detection outcome.

## D.1 INTERLEAVE-VLA ON $\pi_0$

We make minimal architectural changes to the $\pi_0$ (Black et al., 2024) model: only the input processor. Specifically, to enable interleaved image-text instructions, we extend its tokenizer vocabulary by introducing special tokens `<BOI>` (beginning of image) and `<EOI>` (end of image). These newly added tokens are used to delineate image embeddings within the instruction sequence. Specifically, the input tokens are constructed as follows:

```
<BOI> <image>₁...<image>₂₅₆ <EOI> <text> <BOI> <image>₂₅₇...<image>₅₁₂
<EOI> <text> <BOI> <image>₅₁₃...<image>₇₆₈ <EOI> <text> ...
```

Here, each `<image>` token represents a patch embedding from the visual encoder, and the `<BOI>` and `<EOI>` tokens mark the boundaries of each interleaved image segment. This design allows the model to flexibly process multimodal instructions by alternating between image and text tokens within a unified sequence.

Our Interleave-VLA approach is both *effective* and *model-agnostic*, requiring only *minimal modifications*. Its *effectiveness* is evidenced by substantial improvements in generalization performance over $\pi_0$, achieving 2–3× gains as shown in Table 2 and Table 3. Interleave-VLA is *model-agnostic*, seamlessly integrating into existing VLA models without requiring assumptions about the VLM. In Interleave-VLA based on $\pi_0$, the VLM backbone Paligemma (Beyer et al., 2024) demonstrates compatibility despite not being pre-trained on Internet-scale interleaved image-text data. Moreover, our approach introduces only *minimal modifications*, with no architectural changes needed for the underlying VLM backbone. These facts highlight the practicality and broad applicability of Interleave-VLA for advancing multimodal robot learning.

## D.2 INTERLEAVE-VLA ON OPENVLA

While architectural changes are not required to the VLM backbone—as demonstrated in our adaptation from $\pi_0$—we further investigate whether modifying the backbone architecture affects its effectiveness. Specifically, we replace OpenVLA's original Prismatic VLM (Karamcheti et al., 2024) backbone with InternVL2.5 (Chen et al., 2024b), which inherently supports the interleaved image-text format. As shown in Figure 6, our Interleave-VLA adaptation based on OpenVLA continues to function effectively, achieving more than double the performance of the original OpenVLA. This result further highlights the model-agnostic nature of Interleave-VLA and its compatibility with diverse VLA architectures. We have tested on different VLM backbone for OpenVLA in Table 11 and found that changing OpenVLA's VLM backbone has negligible effect on performance.

## D.3 INTERLEAVE-VLA INFERENCE SPEED

To clarify the computational overhead introduced by Interleave-VLA during inference, we note that although the attention cost scales quadratically with the number of input images, the corresponding coefficient is small in practice. Consequently, the additional latency remains modest for typical real-world settings where the number of images is fewer than five. Empirical benchmarks conducted on an RTX 4090 GPU using our implementation of $\pi_0$ show that the inference latency is well approximated by:

$$t = 1.2n^2 + 1.5n + 221, \tag{1}$$

indicating that the dominant term is effectively constant. When $n < 5$, the latency increase stays below 50 ms. Figure 9 reports the measured latency and percentage increase for different numbers of images, confirming that the overhead remains limited under common usage patterns.

In all tasks studied in this work, such latency differences do not affect control performance. For highly dexterous or fast dynamic settings where even small delays may influence throughput, the impact can be mitigated via Real-Time Control (RTC) mechanisms as demonstrated in prior work Black et al. (2025). Furthermore, the overhead can be reduced by employing high-speed image tokenizers (e.g., Vasu et al. (2023)), as the image tokens are primarily used to encode semantic content rather than fine-grained spatial detail.

| Method (VLM, Input) | L1 | L2 | L3 |
|---|---|---|---|
| Interleave-VLA (InternVL2.5, Interleaved) | **83.14** | **58.14** | **64.00** |
| OpenVLA (Prismatic, Text) | 53.71 | 23.00 | 23.86 |
| OpenVLA (InternVL2.5, Text) | 45.29 | 24.71 | 29.71 |

Table 11: Comparing Interleave-VLA and OpenVLA with different VLM backbones on VIMA-Bench.

## E RELIABILITY ANALYSIS OF THE INTERLEAVED DATASET GENERATION PIPELINE

Figure 10 illustrates the two *complementary* stages of our generation pipeline: Owlv2 and QwenVL+SAM. Empirical observations indicate that QwenVL+SAM excels at handling open-world objects, such as the green star shown in the top right of the figure. However, it struggles

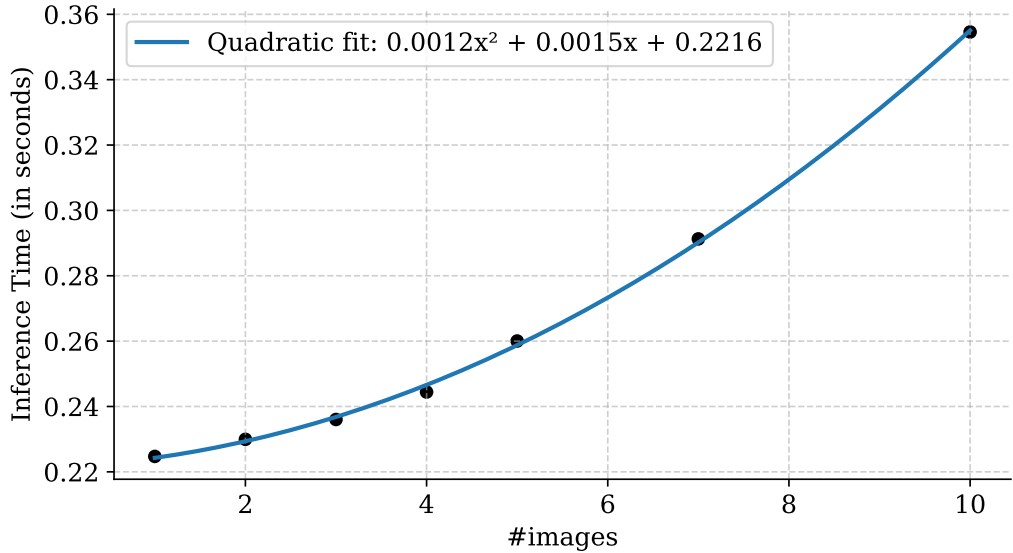

Figure 9: **Interleave-VLA Inference time w.r.t number of images.** When number of images is 1 – 2, it is typically the cost of Text-VLA model. Interleave-VLA takes in more images because of interleaving them in instruction. While the inference cost scales quadratically with the number of input images, the coefficient is very small compared to the constant term. As is typical in most tasks (usually image number under 5), such modest latency increases do not incur performance.

in cluttered scenes or under occlusions, as depicted on the left side of the figure, where Owlv2 demonstrates superior performance. Notably, the combined approach significantly reduces failure rates, although both methods face challenges under severe occlusions or low image resolution.

To evaluate accuracy, we randomly sampled 200 examples from the generated dataset and verified whether the detected images matched the corresponding text. Each sample may contain multiple key objects, and we considered it a failure if any key object was not detected. The individual error rates for QwenVL+SAM and Owlv2 were 22.1% and 17.4%, respectively, while the combined approach reduced the error rate to just *4.4%*. These results highlight the effectiveness of integrating these two models to enhance the reliability of the generation pipeline.

We then expanded our analysis to a larger set of 400 cases and categorized the dominant failure modes at each stage of the data-construction pipeline, ranging from LLM command parsing to object-detection grounding. As shown in Figure 7, the most frequent issues arise from visual grounding errors, such as wrong object identification, missed detections, or incomplete bounding boxes, followed by command-level ambiguities and parsing mismatches. These systematic error patterns highlight (1) the existing robot dataset is not perfect in annotation and (2) both LLM parsing and visual detection modules could be further improved. For additional qualitative examples illustrating these failure categories, please refer to Table 10.

## F    HALLUCINATION ANALYSIS OF VLA MODELS

While Figure 5 provides a qualitative examination of VLA hallucination patterns as revealed through attention visualizations, we further conduct a rigorous quantitative analysis of hallucinations in the SimplerEnv-Bridge experiment. Existing hallucination-evaluation methodologies designed for vision-language models (VLMs) do not directly transfer to vision-language-action (VLA) models, as VLAs do not generate textual outputs but instead execute action trajectories.

To address this gap, we manually inspected and categorized hallucination failure modes across all rollouts. These failure modes are divided into high-level and low-level categories. High-level failures include *Jitter*, where the VLA fails to infer the task intention and the robot arm jitters or drifts,

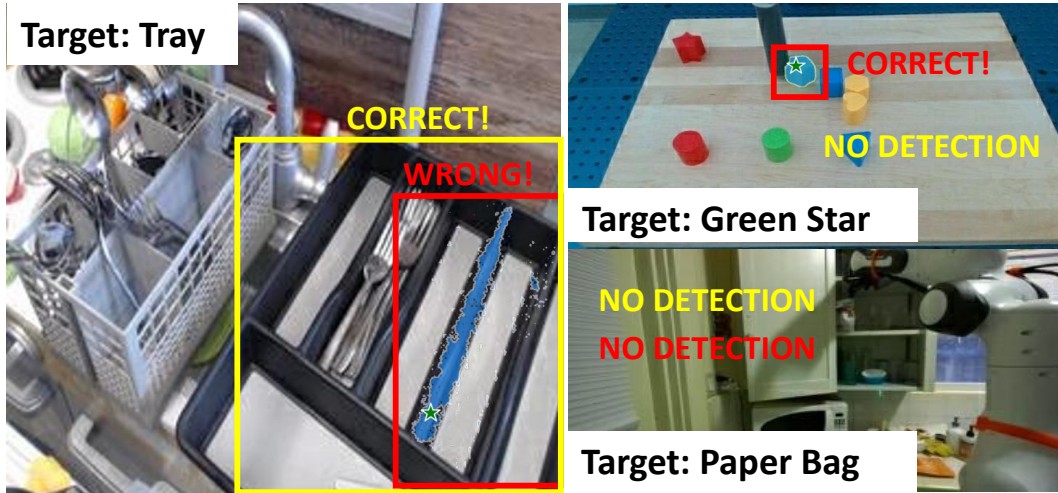

Figure 10: Red: QwenVL+SAM and Yellow: Owlv2. Individual error rates are 22.1% and 17.4%, respectively. The combined error rate is reduced to *4.4%*.

and *Wrong Object*, where the model confidently selects an incorrect object. Low-level failures consist of *Grasp Failed* and *Place Failed*, corresponding to execution-level errors. Each rollout is assigned a single failure category, as the categories are mutually exclusive.

We evaluate the two baselines introduced in Table 2: $\pi_0$ (Text-VLA) and $\pi_0$ (Interleave-VLA full). The aggregated results are presented in Figure 11 and Table 12. The comparisons show that Interleave-VLA substantially reduces high-level hallucinations relative to Text-VLA, shifting most of its errors from intention-level misunderstandings to lower-level execution failures.

Looking ahead, further progress in hallucination mitigation for VLA systems will likely require automatic and scalable methods for quantifying hallucination rates in action trajectories.

| Failure Mode | Model | In-domain (%) | Visual (%) | Novel Object (%) | Novel Category (%) |
|---|---|---|---|---|---|
| Jitter | Interleave-VLA | 0.0 | 0.0 | 0.0 | 3.4 |
| | Text-VLA | 0.0 | 0.0 | 6.5 | 27.5 |
| Wrong Object | Interleave-VLA | 0.0 | 0.0 | 15.3 | 4.4 |
| | Text-VLA | 0.0 | 0.0 | 47.4 | 36.0 |
| Grasp Failed | Interleave-VLA | 21.5 | 22.9 | 16.2 | 38.1 |
| | Text-VLA | 21.8 | 26.5 | 9.3 | 15.5 |
| Place Failed | Interleave-VLA | 7.3 | 3.6 | 14.6 | 0.0 |
| | Text-VLA | 9.0 | 2.1 | 6.0 | 0.0 |
| Total | Interleave-VLA | **28.8** | **26.6** | **46.1** | **45.8** |
| | Text-VLA | **30.8** | **28.6** | **69.2** | **79.0** |

Table 12: Detailed breakdown of hallucination failure modes for $\pi_0$ (Text-VLA) and Interleave-VLA across all task categories. The table reports the per-scenario failure rates for each failure mode: high-level intention errors (*Jitter*, *Wrong Object*) and low-level execution errors (*Grasp Failed*, *Place Failed*). Interleave-VLA exhibits substantially lower high-level hallucinations, particularly in out-of-domain settings (Novel Object, Novel Category), while most remaining errors arise from low-level action execution. In contrast, Text-VLA shows markedly higher high-level intention failures, leading to increased overall hallucination rates. These numeric results correspond to the aggregated visualization shown in Figure 11.

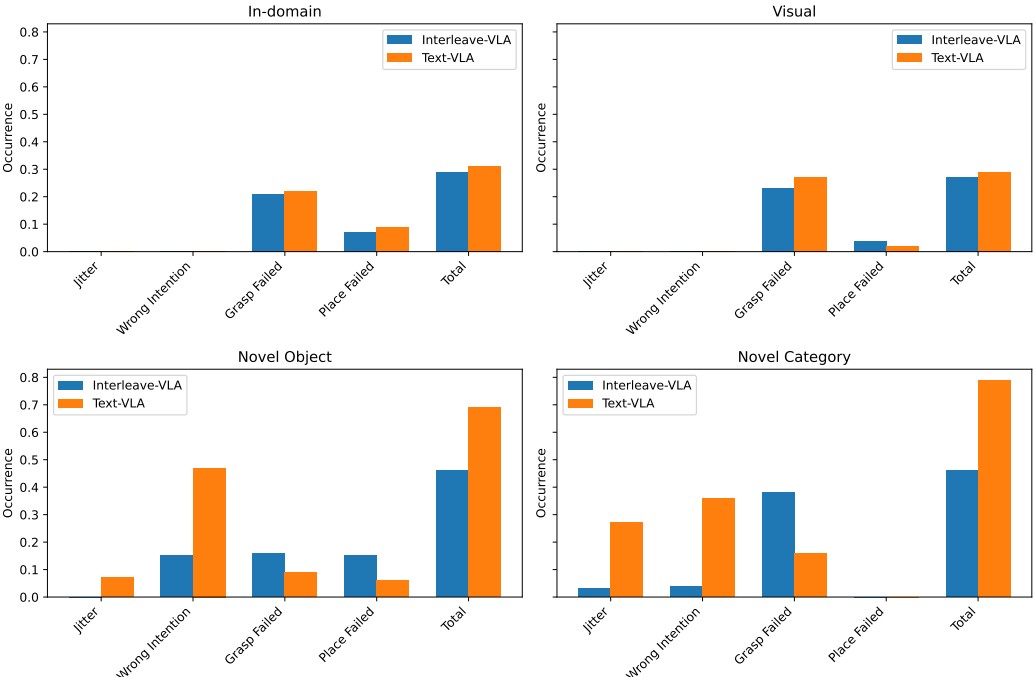

Figure 11: Quantitative hallucination analysis of $\pi_0$ with text-only instructions (Text-VLA) and interleaved image–text instructions (Interleave-VLA). Across all task categories, Interleave-VLA achieves higher overall success rates. Each failed rollout is attributed to a single failure mode: high-level intention errors (*Jitter*, *Wrong Intention*) or low-level execution errors (*Grasp Failed*, *Place Failed*). Interleave-VLA substantially reduces high-level hallucinations, with most residual failures arising from low-level action generation. In contrast, Text-VLA exhibits significantly more high-level intention errors, particularly in out-of-domain scenarios, leading to reduced overall success.

# G  EVALUATION DETAILS

## G.1  EVALUATION ON SIMPLERENV

### G.1.1  SIMPLERENV EVALUATION TASKS

Our evaluation on SimplerEnv (Li et al., 2024) includes both In-Domain and Out-of-Domain tasks. The In-Domain tasks follow the original SimplerEnv WidowX BridgeData V2 Visual Matching setup. Since SimplerEnv tasks use text-based instructions, we adapt them into interleaved image-text instructions using the method described in Section 3.3, based on the first frame of the rollout before the robot arm begins moving.

In the WidowX BridgeData V2 setup, SimplerEnv does not support generalization tasks (referred to as the Variant Aggregation setup). To overcome this limitation, we introduce a set of challenging Out-of-Domain tasks inspired by the Open Vocabulary manipulation evaluations (Stone et al., 2023). Unlike prior methods that rely on separate VLMs to detect target objects in the scene and inject this information into the robot policy, our Interleave-VLA directly leverages interleaved image-text instruction to perform these tasks without requiring additional modules. These tasks are deliberately designed to be more challenging than the original SimplerEnv tasks, requiring the robot to generalize to novel objects and environments unseen during training on BridgeData V2 (Walke et al., 2023a).

We describe the 13 tasks (4 In-Domain and 9 Out-of-Domain, as illustrated on the left of Figure 4) used in the SimplerEnv evaluation. The Out-of-Domain tasks are introduced in the order they appear from top left to bottom right, in Figure 4.

1. **widowx spoon on towel** (In-Domain): This task is part of the original SimplerEnv Visual Matching setting and is included in the BridgeData V2.

2. **widowx carrot on plate** (In-Domain): Also from the original SimplerEnv Visual Matching setting, this scenario is present in the training data.

3. **widowx stack cube** (In-Domain): This stacking task is included in the original SimplerEnv Visual Matching setting and present in the training data.

4. **widowx put eggplant in basket** (In-Domain): This task is part of the original SimplerEnv Visual Matching setting and is present in the training data.

5. **widowx spoon on towel, unseen environment** (Out-of-Domain, Visual Generalization): The environment overlay is sourced from the RT-1 Dataset (Brohan et al., 2022) and is not seen during Bridge V2 training. The robot must generalize to a novel environment.

6. **widowx spoon on towel, unseen tablecloth** (Out-of-Domain, Visual Generalization): The tablecloth overlay is a random image from the internet, unseen in Bridge V2 training data, requiring the robot to generalize to new visual backgrounds.

7. **widowx spoon on towel, unseen lighting** (Out-of-Domain, Visual Generalization): The scene lighting changes dynamically with different colors (RGB) at 5Hz. The robot must generalize to novel and rapidly changing lighting conditions.

8. **widowx redbull on plate** (Out-of-Domain, Semantic Generalization): This is an unseen object from a known category. While similar cans (e.g., tomato can) appear in training, the Redbull can is new. The robot must use language grounding to identify and manipulate the correct object among distractors (e.g., a Coca-Cola can).

9. **widowx tennis ball in basket** (Out-of-Domain, Semantic Generalization): This is an unseen object from a known category. While similar balls (e.g., white ball, blue ball) appear in training, the tennis ball is new. The robot must use language grounding to select and manipulate the correct object among distractors (an orange and a ping pong ball).

10. **widowx zucchini on plate** (Out-of-Domain, Semantic Generalization): This task involves an unseen object from a known category. While a similar zucchini appears only once among 40,000 training episodes, this specific zucchini is entirely novel. The robot must leverage language grounding to accurately identify and manipulate the correct object, distinguishing it from distractors such as a carrot.

11. **widowx zucchini on saucer dish** (Out-of-Domain, Semantic Generalization): This task introduces a novel zucchini instance and an unfamiliar destination—a saucer dish—both of which

are unseen objects from known categories. The robot must ground the instruction to correctly identify the target zucchini and place it onto the saucer, discriminating it from distractors such as a carrot and a regular plate.

12. **widowx toy dinosaur on towel** (Out-of-Domain, Semantic Generalization): This is a completely unseen category. The robot must use language grounding to identify and manipulate the correct object among distractors (a toy elephant).

13. **widowx tape measure in basket** (Out-of-Domain, Semantic Generalization): This is a completely unseen category. The robot must use language grounding to identify and manipulate the correct object among distractors (a purple eggplant).

14. **widowx stapler on paper pile** (Out-of-Domain, Semantic Generalization): This task involves a completely unseen category for both the object and the destination. The robot must leverage language grounding to accurately identify and manipulate the correct object (a stapler) among distractors (e.g., a spatula) and place it onto the unseen destination, the paper pile.

### G.1.2 SIMPLERENV BASELINES

Our experiment in Table 2 compares Interleave-VLA (adapted from $\pi_0$) with $\pi_0$ (Black et al., 2024), RT-1-X (Brohan et al., 2022), and Octo-Base (Team et al., 2024). RT-1-X and Octo models are evaluated using their official checkpoints and code, following the evaluation protocol in the SimplerEnv (Li et al., 2024) repository. For $\pi_0$, we use the reimplementation from the GitHub repository (Zren, 2025), which is specifically trained on BridgeData V2 (Walke et al., 2023a) and supports direct evaluation on SimplerEnv. Interleave-VLA is built upon this reimplemented $\pi_0$ codebase, with modifications to the input tokens and training on the interleaved BridgeData V2, using the interleaved dataset construction pipeline described in Section 3.3. To further highlight the benefits of large-scale, diverse, cross-embodiment data, we also co-train Interleave-VLA with our curated Open Interleaved X-Embodiment Dataset, as detailed in Section 3.3.

Both Interleave-VLA (including the co-trained variant) and $\pi_0$ models were trained with a learning rate of 5e-5, a global batch size of 1024, for approximately 30 epochs. The model input consists of a single observation image (no history), interleaved image-text instruction tokens, one proprioceptive token (no history), and four action tokens. Training takes roughly 2 days on 4×H100 GPUs with a per device batch size of 16. Actions and proprioception across the diverse datasets are normalized to the 7D format: xyz position, Euler orientation, and gripper state, with all values scaled to the range $[-1, 1]$.

The results presented in Table 2 reflect the best performance across checkpoints. Notably, performance can vary significantly between checkpoints, even among those that appear mostly converged. This variability is particularly pronounced for challenging tasks requiring precise manipulation, such as "widowx stack cube". These observations align with findings reported in the $\pi_0$ reimplementation GitHub repository (Zren, 2025).

### G.1.3 SIMPLERENV EVALUATION RESULTS

Table 13 provides detailed generalization results for the top-performing models: $\pi_0$, Interleave-VLA (adapted from $\pi_0$), and Interleave-VLA co-trained, as reported in Table 2. Interleave-VLA consistently surpasses $\pi_0$ across all Out-of-Domain generalization tasks, demonstrating the effectiveness of multimodal learning from interleaved image-text data for both visual and semantic generalization. The co-trained Interleave-VLA model achieves further improvements, especially on semantic generalization tasks such as "RedBull on Plate," where similar RedBull cans are present in the RT-1 dataset for the Google robot. This highlights positive cross-embodiment task transfer to the WidowX robot. Overall, these results show that training with large-scale, diverse robot data enhances model generalization to novel tasks and robot embodiments, supporting our approach of curating the Open Interleaved X-Embodiment Dataset.

Note that the Unseen Environment setting is omitted for the Interleave-VLA co-trained model because the scene overlay is sourced from the RT-1 Google Robot dataset, which is included in the co-train data. As a result, the model tends to generate actions intended for the Google Robot. During evaluation, however, the robot used is WidowX, leading to a mismatch in embodiment and causing the model to produce incorrect actions.

Table 13: Detailed evaluation results on 9 Out-of-Domain generalization tasks based on SimplerEnv. Success rates (%) are reported for $\pi_0$, Interleave-VLA (adapted from $\pi_0$), and Interleave-VLA co-trained with our Open Interleaved X-Embodiment Dataset, covering both visual and semantic generalization. Generalization results confirm that Interleave-VLA outperforms $\pi_0$ across all tasks, with further cross-embodiment improvements from co-training.

| Model | Visual Generalization | | | Semantic Generalization | | | | | | Average |
|---|---|---|---|---|---|---|---|---|---|---|
| | Unseen Tablecloth | Unseen Environment | Unseen Lighting | Redbull on Plate | Tennis Ball in Basket | Zucchini on Plate | Toy Dinosaur on Towel | Tape Measure in Basket | Stapler on Paper Pile | |
| $\pi_0$ | 78.0 | 77.0 | 59.2 | 0.0 | 30.0 | 50.0 | 24.0 | 1.0 | 38.0 | 39.7 |
| Interleave-VLA | **80.0** | **79.0** | **61.3** | **35.0** | **73.0** | **83.0** | **39.0** | **53.0** | **70.0** | **63.4** |

## G.2 EVALUATION ON VIMA-BENCH

### G.2.1 VIMA-BENCH EVALUATION TASKS

We evaluate performance on the majority of VIMA-Bench tasks, but excluding those requiring historical memory. Memory-dependent tasks are omitted because Interleave-VLA, like common VLA models (Kim et al., 2024; O'Neill et al., 2024; Brohan et al., 2022; 2023; Black et al., 2024; Team et al., 2025; Fang et al., 2025a; Wen et al., 2025), is designed for memory-independent, first-order Markov settings. In general, common VLA models characterize the conditional distribution $p(\mathbf{A}_t|\mathbf{o}_t)$, where $\mathbf{A}_t = [\mathbf{a}_t, \mathbf{a}_{t+1}, \ldots, \mathbf{a}_{t+H-1}]$ represents a sequence of future actions, and $\mathbf{o}_t$ denotes the current observation (comprising multiple RGB images, a language command, and the robot's proprioceptive state). Extending VLAs to handle historical memory in interleaved instruction scenarios remains an interesting direction for future work.

VIMA-Bench employs interleaved image-text instructions for task specification. To evaluate text-instructed VLA models, we transform these interleaved instructions into text-only instructions by utilizing the shape and texture names provided in the VIMA-Bench codebase. For example:

```
VIMA-Bench Instruction:  Put the        into the     .
Transformed Instruction:  Put the rainbow triangle into the blue
square.
```

### G.2.2 VIMA-BENCH BASELINES

We evaluate Interleave-VLA (adapted from OpenVLA) against several baselines: OpenVLA (Kim et al., 2024), VIMA-Gato (Jiang et al., 2023), VIMA-Flamingo (Jiang et al., 2023), and VIMA-GPT (Jiang et al., 2023). All models are trained on the same dataset generated using an oracle model, which has access to the exact 2D poses of all objects in the scene. This dataset generation process is provided by VIMA. For OpenVLA, the training data consists of text-instructed samples. Both Interleave-VLA and OpenVLA are trained on an equivalent amount of the generated VIMA dataset using the following training hyperparameters: a constant learning rate of 2e-5 and a global batch size of 128. This comparison demonstrates the effectiveness of Interleave-VLA in improving generalization performance over existing VLA models. The results for VIMA-Gato, VIMA-Flamingo, and VIMA-GPT are taken from the original VIMA paper (Jiang et al., 2023) and serve as additional benchmarks. These models, adapted by the VIMA team, serve as benchmarks to assess the progression of VLA models from earlier architectures like Gato, Flamingo, and GPT to the more advanced OpenVLA.

### G.2.3 VIMA-BENCH EVALUATION RESULTS

The detailed results for the memory-independent VIMA-Bench tasks are presented in Table 14. The results demonstrate that Interleave-VLA benefits significantly from interleaved image-text instructions, which enhance its ability to identify and manipulate the correct object by $2\times$. This approach proves more effective than relying solely on text descriptions to distinguish objects with the desired texture and shape among distractors.

Table 14: Detailed VIMA-Bench results for L1, L2, and L3 level generalization evaluations. Interleave-VLA generally outperforms other VLA models and improves the generalization capacity of OpenVLA (Kim et al., 2024) by over $2\times$.

| Model Name | task1 | task2 | task3 | task4 | task7 | task11 | task15 | AVG |
|---|---|---|---|---|---|---|---|---|
| **VIMA-Bench L1** | | | | | | | | |
| OpenVLA (Kim et al., 2024) | 83 | 70 | 78 | 4 | **92** | 0 | 49 | 53.71 |
| Interleave-VLA | **87** | **82** | 81 | 54 | 82 | **100** | **96** | **83.14** |
| VIMA-Gato | 79 | 68 | **91** | **57** | 74 | 61 | 83 | 73.29 |
| VIMA-Flamingo | 56 | 58 | 63 | 48 | 62 | 66 | 40 | 56.14 |
| VIMA-GPT | 62 | 57 | 41 | 55 | 54 | 77 | 41 | 55.29 |
| **VIMA-Bench L2** | | | | | | | | |
| OpenVLA (Kim et al., 2024) | 18 | 20 | 68 | 2 | 31 | 0 | 22 | 23.00 |
| Interleave-VLA | 36 | 32 | 75 | 44 | 26 | **100** | **94** | 58.14 |
| VIMA-Gato | **56.5** | **53.5** | **88** | **55.5** | 53 | 63 | 81.5 | **64.43** |
| VIMA-Flamingo | 51 | 52.5 | 61.5 | 49.5 | **55.5** | 82 | 42 | 56.29 |
| VIMA-GPT | 52 | 52 | 49.5 | 54.5 | 51 | 76.5 | 43 | 54.07 |
| **VIMA-Bench L3** | | | | | | | | |
| OpenVLA (Kim et al., 2024) | 27 | 36 | 61 | 3 | 26 | 0 | 14 | 23.86 |
| Interleave-VLA | **52** | 55 | 81 | 53 | 46 | **98** | 63 | **64.00** |
| VIMA-Gato (Jiang et al., 2023) | 51 | **58** | **84.5** | **56.5** | 49 | 65 | 52 | 59.43 |
| VIMA-Flamingo (Jiang et al., 2023) | 49 | 50 | 66.5 | 47 | 50 | 66 | 30.5 | 51.29 |
| VIMA-GPT (Jiang et al., 2023) | **52** | 51 | 55 | 49.5 | **50.5** | 82 | 37 | 53.86 |

## G.3 EVALUATION ON REAL ROBOT

### G.3.1 REAL ROBOT EVALUATION TASKS

We evaluate on two distinct manipulation tasks: Lift and Pick&Place, corresponding to the first and second rows of results shown in Table 3. Visual illustrations of these tasks are shown on the right side of Figure 4. The tasks are designed to be challenging, requiring the robot to generalize to novel objects not seen during training. We describe these tasks in more detail.

The Lift task includes:

1. **Lift pepper** (In-Domain): 20 demonstrations collected with varied object arrangements and positions.

2. **Lift cup** (In-Domain): 20 demonstrations collected with varied object arrangements and positions.

3. **Lift corn** (In-Domain): 20 demonstrations collected with varied object arrangements and positions.

4. **Lift lemon** (Out-of-Domain, Semantic Generalization): The target is an unseen object, as lemons are not included in the collected demonstrations. Although the lemon category appears in the pretraining data, it appears with different textures, robots, and environments. VLA models must utilize language grounding to accurately identify and lift the target lemon among two distractor items.

5. **Lift bean** (Out-of-Domain, Semantic Generalization): The target belongs to a completely unseen category, as beans are absent from both the collected demonstrations and the pretraining dataset. VLA models must rely on language grounding to correctly identify and lift the target bean among two distractor items.

6. **Lift spoon** (Out-of-Domain, Semantic Generalization): The target is an unseen object from a known category, as the demonstrations do not include this specific spoon. While the spoon category appears in the pretraining data, it is represented with different textures, robots, and environ-

ments. VLA models must leverage language grounding to accurately identify and lift the target spoon among two distractor items.

The Pick&Place task includes:

1. **Pick up kitchen cutter and place into the pot** (In-Domain): 20 demonstrations collected with varied object arrangements and positions.

2. **Pick up ladle and place into the pot** (In-Domain): 20 demonstrations collected with varied object arrangements and positions.

3. **Pick up pasta server and place into the pot** (In-Domain): 20 demonstrations collected with varied object arrangements and positions.

4. **Pick up the white and blue spatula and place it into the pot** (Out-of-Domain, Semantic Generalization): The target is an unseen object from a known category. The demonstrations do not include any spatula. While the spatula category appears in the pretraining data, it is shown with different textures, robots, and environments. VLA models must utilize language grounding to accurately identify and manipulate the target spatula among two distractor kitchenware items.

5. **Pick up the black and white spatula and place it into the pot** (Out-of-Domain, Semantic Generalization): Similar to the previous task, but the target spatula is black and white. The robot must leverage language grounding to correctly identify and manipulate the target spatula among two distractor kitchenware items.

### G.3.2 REAL ROBOT BASELINES

We compare Interleave-VLA (adapted from $\pi_0$) with pretraining against the following baselines: $\pi_0$ with pretraining and Interleave-VLA without pretraining. The pretraining dataset is a subset of our curated Open Interleaved X-Embodiment Dataset, as described in Section 3.3. Interleave-VLA w/ PT is pretrained on this dataset and subsequently fine-tuned on the collected demonstrations from the FANUC robot arm before evaluation. For $\pi_0$ w/ PT, the same pretraining and fine-tuning protocol is applied, except the dataset is not interleaved. This setup allows for a direct comparison to evaluate the benefits of interleaved image-text instructions for generalization. The Interleave-VLA w/o PT is trained exclusively on the collected FANUC demonstrations, without exposure to the Open Interleaved X-Embodiment Dataset, enabling us to assess the impact of large-scale, diverse pretraining on performance. All models are fine-tuned with a learning rate of 5e-5, a global batch size of 128, and evaluated across several checkpoints to mitigate the performance variability noted in Appendix G.1.2.

### G.3.3 REAL ROBOT EVALUATION RESULTS

Tables 15 and 16 present the detailed evaluation results for the Lift and Pick&Place tasks, respectively. Interleave-VLA, adapted from $\pi_0$, is compared against $\pi_0$ and Interleave-VLA without pretraining (w/o PT). In generalization tasks, Interleave-VLA consistently outperforms $\pi_0$ in semantic generalization by $2\times$, highlighting the effectiveness of multimodal learning from interleaved image-text data. The results further demonstrate that pretraining on the Open Interleaved X-Embodiment Dataset significantly enhances performance across all tasks. For small-scale datasets (60 demonstrations in total per task), pretraining on the Open Interleaved X-Embodiment Dataset proves essential for achieving strong performance, as cross-embodiment pretraining enables the model to learn more robust representations and generalize effectively, even to the FANUC robot, which is not included in the pretraining data.

## H SCALABILITY OF INTERLEAVE-VLA WITH THE OPEN INTERLEAVED X-EMBODIMENT DATASET

The Open Interleaved X-Embodiment Dataset, detailed in Section 3.3, empowers Interleave-VLA to scale efficiently with increasing data. This section demonstrates the scalability of Interleave-VLA through pretraining and co-training strategies in varying data regimes.

**Pretraining for Low-Data Regimes:** As shown in Table 3, pretraining on the curated Open Interleaved X-Embodiment Dataset is essential for achieving strong performance on real robot tasks.

Table 15: Detailed evaluation of the "Lift task". We conduct 12 trials for each object and report both the number of successful trials (# Succ) and the number of trials where the correct object is manipulated (# Acc).

| Category | Task | # Trials | Interleave-VLA w/ PT # Succ / # Acc | Interleave-VLA w/o PT # Succ / # Acc | $\pi_0$ w/ PT # Succ / # Acc |
|---|---|---|---|---|---|
| In-Domain | pepper | 12 | 7/12 | 2/4 | 7/10 |
| In-Domain | corn | 12 | 9/12 | 0/4 | 4/12 |
| In-Domain | cup | 12 | 8/12 | 0/4 | 3/12 |
| Out-of-Domain | spoon | 12 | 9/11 | 0/2 | 9/11 |
| Out-of-Domain | bean | 12 | 9/12 | 0/4 | 1/1 |
| Out-of-Domain | lemon | 12 | 8/12 | 0/4 | 2/5 |
| | Mean Success / Accuracy Rate | | 69.4 % / 98.6 % | 2.8 % / 30.6 % | 36.1 % / 70.8 % |

Table 16: Detailed evaluation on "Pick&Place task". We conduct 12 trials for each object and report both the number of successful trials (# Succ) and the number of trials where the correct object is manipulated (# Acc).

| Category | Task | # Trials | Interleave-VLA w/ PT # Succ / # Acc | Interleave-VLA w/o PT # Succ / # Acc | $\pi_0$ w/ PT # Succ / # Acc |
|---|---|---|---|---|---|
| In-Domain | pasta server | 12 | 6/8 | 4/8 | 7/10 |
| In-Domain | spoon | 12 | 7/10 | 1/7 | 7/9 |
| In-Domain | knife | 12 | 4/7 | 2/7 | 4/12 |
| Out-of-Domain | spatula | 12 | 3/8 | 0/8 | 1/1 |
| Out-of-Domain | black spatula | 12 | 6/8 | 0/6 | 4/5 |
| | Mean Success / Accuracy Rate | | 43.3 % / 68.3 % | 11.7 % / 60 % | 38.3 % / 61.7 % |

This is particularly important due to the limited size of the FANUC dataset, which contains only 60 demonstrations per task. Pretraining on the significantly larger and more diverse Open Interleaved X-Embodiment Dataset enables Interleave-VLA to learn robust representations that generalize effectively to the FANUC robot, even though it is not included in the pretraining data.

**Co-Training for High-Data Regimes:** Co-training with additional datasets from the Open Interleaved X-Embodiment Dataset further enhances performance in semantic generalization tasks. While the Bridge Dataset V2 is already extensive and diverse, making substantial improvements challenging, co-training yields additional gains in semantic generalization. This demonstrates that interleaved training facilitates cross-embodiment skill transfer. Detailed results are presented in Table 17.

Table 17: Scalability of Interleave-VLA through co-training on the Open Interleaved X-Embodiment Dataset, evaluated under the **SimplerEnv** Out-of-Domain setting. Incorporating datasets beyond Bridge Data V2 in the Open Interleaved X-Embodiment Dataset further improves performance in semantic generalization tasks. The **bold** and underlined values represent the highest and second-highest scores, respectively.

| Base Model | Paradigm | Co-trained | Visual | Novel Object | Novel Category | Avg. |
|---|---|---|---|---|---|---|
| $\pi_0$ (Black et al., 2024) | Interleave-VLA | ✗ | **73.4** | 63.7 | 53.0 | 63.4 |
| $\pi_0$ (Black et al., 2024) | Interleave-VLA | ✓ | 71.5 | **70.7** | **57.3** | **66.5** |

## I  TASK FLEXIBILITY AND EMERGENT GENERALIZATION DETAILS

To highlight the task flexibility and emergent generalization capabilities of Interleave-VLA when faced with unseen instructions, we leverage the interleaved image-text interface to evaluate its performance across diverse user input styles during deployment. The Interleave-VLA model used in this evaluation is directly taken from the SimplerEnv evaluation suite (Table 2 and Table 13) without any additional fine-tuning. A summary of Interleave-VLA's performance statistics is presented in Table 4.

Below, we describe the three tasks and their corresponding prompts in the order they appear in Table 4:

1. **Place {eggplant, carrot} on the plate**. Two types of instructions are provided. The first row includes a hand-drawn sketch of an eggplant and a carrot, created by a human on-the-fly. The second row features a sketch-style image of an eggplant and a carrot sourced from the Internet.

2. **Place {green, yellow} block on the towel**. Two types of instructions are included. The first row contains a hand-drawn sketch of a green and yellow block, created by a human on-the-fly. The second row features random images representing a green and yellow block, sourced from the Internet.

3. **Place {block, spoon} on the towel**. Two types of instructions are used. The first row includes a hand-drawn sketch of a block and a spoon, created by a human on-the-fly. The second row features cropped images of the desired target objects, captured from a screen by a human on-the-fly.

Interleave-VLA demonstrates remarkable emergent generalization capabilities, even when faced with diverse instruction styles such as Internet images, object crops (from a familiar input style but with unseen images), and sketches (a completely novel input style not encountered during training). These emergent capabilities go beyond the typical generalization to novel objects and environments evaluated in prior VLA models (Black et al., 2024; Kim et al., 2024). They highlight Interleave-VLA's adaptability to new tasks and instruction formats, showcasing its practical flexibility in processing diverse multimodal inputs.

## J    OPEN INTERLEAVED X-EMBODIMENT DATASET DETAILS

The Open Interleaved X-Embodiment Dataset, curated as described in Section 3.3 for training Interleave-VLA, integrates data from 11 sources within the Open X-Embodiment Dataset. To ensure coherent training and facilitate cross-embodiment transfer, the action space across all datasets is standardized to a unified 7D pose format: xyz position, Euler orientation, and gripper state. This normalization adheres to practices established in recent VLA research (Kim et al., 2024; Black et al., 2024; Team et al., 2024). Our dataset features an extensive variety of over 3500 diverse object categories, as depicted on the left of Figure 3. Additionally, Figure 12 highlights the wide range of skills encompassed within the dataset and provides a detailed breakdown of its composition and partitioning.

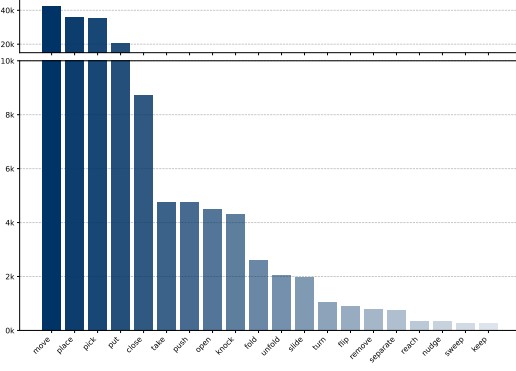

| Interleaved X-Embodiment Dataset Composition | |
| --- | --- |
| RT-1 (Brohan et al., 2022) | 41.01% |
| Bridge (Walke et al., 2023a) | 28.25% |
| BC-Z (Jang et al., 2022) | 20.34% |
| Language Table (Lynch et al., 2023) | 7.81% |
| UTAustin Mutex (Shah et al., 2023) | 0.71% |
| Jaco Play (Dass et al., 2023) | 0.51% |
| Berkeley Autolab UR5 (Chen et al.) | 0.47% |
| IAMLab CMU Pickup Insert (Saxena et al., 2023) | 0.30% |
| Stanford Hydra (Belkhale et al., 2023) | 0.27% |
| UTAustin Sirius (Liu et al., 2023b) | 0.26% |
| UCSD Kitchen (Yan et al., 2023) | 0.07% |

Figure 12: **Left:** Our Open Interleaved X-Embodiment Dataset is diverse in skills. **Right:** Composition of open data sources in our curated Open Interleaved X-Embodiment Dataset.

