# OpenReview forum: "Interleave-VLA: Enhancing Robot Manipulation with Image-Text Interleaved Instructions"
_ICLR.cc/2026/Conference — ICLR 2026 Poster_

### Official Review · Reviewer_9vDg · 2025-10-28

**Soundness:** 3
**Presentation:** 3
**Contribution:** 3
**Rating:** 6
**Confidence:** 4

**Summary:**

This paper presents Interleave-VLA that predicts robotic actions based on multimodal (vision and language) instructions. To train Interleave-VLA, the authors propose a framework that automatically converts the text instructions in the existing large-scale robot learning dataset (Open X-Embodiment) to multimodal instructions. Experiments were conducted on real-world and simulated environments, and results demonstrate that learning from multimodal instructions significantly improves out-of-domain generalization.

**Strengths:**

- [S1] The paper explores a novel direction where VLA models utilize multimodal language instructions, and the proposed approach (Interleave-VLA) is simple and does not require any architectural modifications.
- [S2] The experimental results seem interesting and clearly demonstrate the effectiveness of Interleave-VLA.
- [S3] The paper is well-written and easy to understand. All figures and equations succinctly describe the details how Interleave-VLA works.

**Weaknesses:**

- [W1] When it comes to real-world scenarios, providing VLAs with multimodal instructions seems unnatural and cumbersome. Furthermore, additional complexity is required to convert textual instructions to multimodal instructions if users provide text (or verbal) instructions.
- [W2] The performance of the proposed system is inherently dependent on the generalization capabilities of off-the-shelf detectors, such as OWLv2 and Segment Anything.

**Questions:**

N/A

---

> ### Author Response · Authors · 2025-11-21
>
> **W1:** Overhead of Instruction Conversion.
>
> **A1:** Thank you for raising this concern. We would like to clarify that our method does not require users to convert textual instructions into multimodal ones. As shown in Table 2 of our paper, the interleaved paradigm preserves the VLA’s ability to follow purely textual instructions. Specifically, Interleave-VLA (Partial), which is trained with interleaved inputs but evaluated with text-only inputs, achieves stronger performance compared to original $\pi_0$ with exactly the same test-time input:
>
> | Base Model | Paradigm               | Train/Eval Modality | In-Domain (%)      | Visual (%)          | Novel Object (%)        | Novel Category (%)       |
> |------------|------------------------|---------------------|--------------------|---------------------|-------------------------|--------------------------|
> | $\pi_0$    | Text-VLA               | Text/Text           | $68.1\pm1.3$       | $72.4\pm1.1$        | $26.0\pm3.6$            | $19.3\pm1.5$             |
> | $\pi_0$    | Interleave-VLA (Partial) | Interleave/Text     | $\mathbf{70.1\pm0.9}$ | $\mathbf{76.8\pm0.2}$ | $\mathbf{35.8\pm0.2}$    | $\mathbf{20.9\pm1.9}$    |
>
> However, while Interleave-VLA preserves strong performance under text-only inputs, relying solely on textual instructions can still lead to ambiguity, especially when describing unfamiliar or visually complex objects. Such ambiguity ultimately limits a VLA model’s generalizability. To address this, and to further improve performance on out-of-domain objects or tasks (as demonstrated in our real-world experiments in Table 3), we provide an intuitive GUI interface in practice that facilitates users in constructing multimodal instructions.
>
> **W2:** Bottleneck of Interleave-VLA.
>
> **A2:** Thank you for the thoughtful comment. We agree that the performance of our system is influenced by the generalization capabilities of the off-the-shelf detectors we employ, such as OWLv2 and Segment Anything. Nevertheless, the continuous improvements in these detectors ultimately strengthen the underlying VLA model. Our analysis is as follows:
>
> 1. **Lower Bound:** Even in cases where the detectors fail, the system’s performance remains on par with the base VLA using text-only inputs, as discussed in **A1** and evidenced by Table 2 in our manuscript. This guarantees that the interleaved paradigm does not harm the baseline capabilities.
> 2. **Upper Bound:** The detectors integrated into our pipeline, particularly Qwen2.5-VL, exhibit strong object-grounding performance in practice, often surpassing the base VLM components used in existing VLAs, such as Prismatic for OpenVLA and PaliGemma for $\pi_0$. This enables us to construct the Open Interleaved X-Embodiment Dataset with high-quality, object-centric annotations. Rather than constraining the VLA, these detectors provide stable and reliable supervisory signals that enhance generalization. Furthermore, our method is **orthogonal** to future improvements in foundation models. As these models continue to advance, the performance of our system is expected to scale accordingly.
>
> In summary, Interleave-VLA’s performance is bounded below by the base VLA without interleaving and bounded above by the continually improving capabilities of modern foundation models.

---

### Official Review · Reviewer_tBwW · 2025-10-31

**Soundness:** 4
**Presentation:** 4
**Contribution:** 3
**Rating:** 8
**Confidence:** 5

**Summary:**

This paper introduces Interleave-VLA, a new paradigm for robotic manipulation that enhances existing Vision-Language-Action (VLA) models by enabling them to understand and act on **interleaved image-text instructions**. The authors argue that traditional text-only instructions are often ambiguous and limit generalization, especially for unseen objects. To address this, their paradigm allows for more flexible and less-biased inputs, such as "Put  on ".

To train this new paradigm, the authors developed a novel, automated pipeline to convert the large-scale Open X-Embodiment dataset into an "Open Interleaved X-Embodiment Dataset," comprising 210k real-world episodes. The paper's core contributions are:
1.  The Interleave-VLA paradigm itself, which is presented as a lightweight, model-agnostic adaptation for existing VLA models.
2.  The large-scale interleaved dataset and the automated pipeline to create it.
3.  A comprehensive set of experiments demonstrating that Interleave-VLA achieves 2x stronger out-of-domain (OOD) generalization in both simulation and the real world compared to text-only baselines.
4.  A novel analysis of "attentional hallucinations" in text-only models, which Interleave-VLA mitigates.
5.  Showcasing emergent, zero-shot capabilities, such as following instructions from hand-drawn sketches or web images.

**Strengths:**

This is a strong paper with several significant contributions.

* **Solves an Important and Intuitively Clear Problem:** The paper's premise is compelling and well-motivated. Using a visual example (a crop, a photo, or even a sketch) is an immediately intuitive and powerful solution. The paper effectively translates this intuition into a practical robot learning paradigm.
* **Comprehensive and Rigorous Experimentation:** The empirical validation is a key strength. The authors test their paradigm thoroughly across three distinct and challenging environments:
    1.  **Simulation (SimplerEnv):** They demonstrate a 2x performance gain on semantically out-of-domain tasks over strong baselines like the base $\pi_0$ model.
    2.  **Real-Robot (FANUC arm):** They replicate these findings in the real world, showing a 2-3x improvement in OOD success rates when using their pre-trained model. This real-world validation is critical and very well-executed.
    3.  **VIMA-Bench:** They show the generality of their *paradigm* by adapting it to a *different* VLA model (OpenVLA) and again showing superior performance, reinforcing their "model-agnostic" claim.
* **High-Value Dataset Contribution:** The creation and open-sourcing of the "Open Interleaved X-Embodiment Dataset" (210k episodes) is a substantial contribution to the community. The automated pipeline (Fig. 3) built from modern foundation models (Qwen, OWLv2, SAM 2) to generate this dataset is itself a clever and valuable piece of engineering, achieving high accuracy (95.6%).
* **Excellent Analysis and Insight:** The paper does not just present results; it provides a strong mechanistic explanation for *why* its approach works. The analysis of "attentional hallucination" (categorized as Attentional Bias, Diffused Attention, and Attention Leakage)  is insightful. Figure 5 provides a clear, qualitative "Aha!" moment for the reader, visually grounding the performance claims.
* **Impressive Emergent Capabilities:** Perhaps the most exciting result is the zero-shot generalization to *unseen instruction modalities*. The ability to handle hand-drawn sketches , internet images , and user crops —without any fine-tuning on such data—dramatically expands the practical utility and flexibility of the system. This demonstrates a deeper level of generalization than just handling unseen objects.
* **Clarity and Presentation:** The paper is exceptionally well-written and organized. The figures (especially 1, 3, and 5) are clear, informative, and effectively communicate the core concepts and results.

**Weaknesses:**

The paper is very strong, and the weaknesses are relatively minor or are areas for further strengthening.

* **Quantification of "Lightweight" Adaptation:** The adaptation is described as "minimal" , primarily adding special tokens (`<BOI>`, `<EOL>`) to the tokenizer for the $\pi_0$ model. While this simplicity is a strength, it's surprising that a model like Paligemma, which was *not* pre-trained on interleaved data, can learn this complex in-context visual grounding behavior so effectively from fine-tuning alone. The paper would be strengthened by a deeper analysis of this. Is the model truly "reading" the visual prompt, or is the prompt image just acting as a strong conditioning signal (e.g., a "blue" prior) that the model learns to associate with the text?
* **Computational Cost Not Quantified:** The authors correctly identify a key limitation: longer image token sequences increase computational demands. This is a very practical concern for real-time robotics. However, this cost is not quantified. The review would be more complete if it included a brief analysis of the trade-off: What is the actual increase in inference latency (e.g., ms per action) when using an interleaved prompt (e.g., text + 2 images) compared to the text-only baseline?
* **Analysis of Failure Cases:** The paper thoroughly documents the (many) successes of Interleave-VLA. However, a "Failure Modes" section is missing. When does Interleave-VLA fail? What if the user-provided sketch is highly ambiguous and matches *two* objects in the scene? What if the text ("red") and image (a blue object) are contradictory? Understanding the limitations of the proposed method is just as important as understanding its strengths.

**Questions:**

These questions are intended to clarify points that could strengthen my assessment.

1.  **Probing the Grounding Mechanism:** Regarding the adaptation of $\pi_0$, I am very impressed by the results given the base VLM was not pre-trained for interleaving. Have you run experiments to probe the nature of the learned grounding? For example, what happens in a contradictory scenario: if the text instruction is "pick up the **red** block" but the interleaved image prompt shows a **blue** block? Which instruction (text or image) does the model prioritize, and does this suggest a deeper "comprehension" or a learned attentional bias?
2.  **Quantifying Inference Overhead:** Could you please quantify the computational overhead mentioned in your limitations? Specifically, what is the approximate increase in the number of tokens processed and the corresponding impact on inference latency (e.g., actions per second) when using a typical interleaved instruction (text + 2 images) versus a text-only instruction?
3.  **Robustness of Sketch Generalization:** The zero-shot sketch generalization  is a fantastic result. How robust is this to the quality of the sketch? Could you provide a few examples of failure cases? I am curious about the limits of this capability, for instance, with sketches that are highly abstract, ambiguous, or could plausibly match multiple objects in the scene.
4.  **Impact of Dataset Pipeline Noise:** Your dataset generation pipeline achieves a high 95.6% accuracy. For the 4.4% of failures, what is the primary error mode (e.g., wrong bounding box, wrong object ID)? Have you observed any impact, even minor, from this label noise on the final trained policy's behavior?

---

> ### Author Response · Authors · 2025-11-21
>
> **Q1:** Probing Grounding Mechanism.
>
> **A1:** To investigate the model’s grounding mechanism, we designed a controlled task: "Pick up a [green / yellow] block and place it on the towel". We evaluated five instruction formats, using the “green block” as an example:
>
> 1. **Text:** “Pick up the green block…”
> 2. **Interleave:** “Pick up the [Image of green block]…”
> 3. **Interleave-Aligned:** “Pick up the green block [Image of green block]…”
> 4. **Interleave-Contradict-v1:** The provided image shows the desired object, but the text does not. “Pick up the yellow block [Image of green block]…”
> 5. **Interleave-Contradict-v2:** The provided image is incorrect, but the text is correct. “Pick up the green block [Image of yellow block]…”
>
> The results are summarized below:
>
> | Metrics / Instruction Format                 | Text (%)         | Interleave (%)    | Interleave-Aligned (%) | Interleave-Contradict-v1 (%) | Interleave-Contradict-v2 (%) |
> | -------------------------------------------- | ---------------- | ---------------- | ---------------------- | ---------------------------- | ---------------------------- |
> | **Success Rate** (Yellow / Green)            | 93.8 / 97.9      | 95.8 / 93.8      | 89.6 / 93.8            | 4.2 / 0.0                    | 62.5 / 89.6                  |
> | **Intention Accuracy** (Yellow / Green)      | 100.0 / 100.0    | 100.0 / 100.0    | 100.0 / 100.0          | 0.0 / 0.0                    | 100.0 / 100.0                |
>
> From these comparisons, we observe that Interleave-VLA is able to jointly interpret image and text modalities (first three columns). In the last two columns, it is notable that Interleave-VLA consistently prioritizes textual instructions over visual cues when the two conflict. This behavior is clearly reflected in the *interleave-contradict-v2* setting, where the model achieves 100% intention accuracy, and further in the *interleave-contradict-v1* setting, where the intention accuracy drops to 0%. (The small 4.2% execution success rate in this setting stems from cases in which the model initially places the wrong object on the towel but subsequently corrects its action.) Overall, these results indicate that the model genuinely understands the combined image–text prompt rather than merely overfitting to specific visual patterns in the prompt image, as further supported by the new Tables 9 and 10 in the updated manuscript.
>
>
> In conclusion, Interleave-VLA demonstrates **(1)** a deep comprehension by understanding both textual and visual modalities correctly and simultaneously (first 3 instruction formats), and **(2)** a consistent, thus non-hallucination-driven preference for textual modalities when contradictions occur (last 2 formats). This experiment has been included in Appendix C of the updated paper. Thank you for the insightful suggestion.
>
> **Q2:** Inference Overhead Quantification.
>
> **A2:** Thank you for the insightful comment. Although the attention cost does grow quadratically with the number of input images, the scaling factor is fairly small in practice. As a result, the added latency remains mild as long as the number of images is limited, which is typically under 5 in most real-world applications.
>
> We benchmarked inference on an RTX 4090 GPU using our implementation of $\pi_0$. The measured latency is well-approximated by $t = 1.2n^2 + 3.9n + 224$, indicating that the dominant cost is a constant term. When $n < 5$, the latency difference remains under 50 ms, as shown below:
>
> | # Images in Prompt | 0 ($\pi_0$) |   1     |   2     |   3     |   4     |   6     |   9     |
> |--------------------|-------------|---------|---------|---------|---------|---------|---------|
> | Time (ms)          | 224.72      | 229.95  | 236.02  | 244.42  | 260.03  | 291.28  | 354.59  |
> | Percent Increase   | +0%         | +2.33%  | +5.03%  | +8.77%  | +15.71% | +29.62% | +57.79% |
>
> In the tasks studied in our paper, such modest latency increases do not affect performance. For highly dexterous or fast dynamic control settings where small latency changes may impact throughput, this can be directly mitigated using Real-Time Control (RTC) mechanisms, as demonstrated in [1]. Moreover, the overhead can be further reduced by adopting high-speed image tokenizers such as [2], since the image tokens primarily encode semantics rather than high-resolution spatial detail.
>
> We appreciate the reviewer’s feedback. We have updated the Limitations section accordingly and included the inference-time benchmarking results in Appendix B.3 and Figure 8.

---

> > ### Author Response · Authors · 2025-11-21
> >
> > **Q3:** Robustness of Sketch Generalization.
> >
> > **A3:** Thank you for your question. To further investigate, we used the task constructed in **Q1** and categorized sketches into different styles based on their clarity and detail:
> >
> > 1. **Normal:** A detailed sketch completed by a human in approximately 15 seconds.
> > 2. **OCR:** A simple sketch of a cube or square with "yellow" or "green" as a caption.
> > 3. **Quick:** A rough sketch created in under 5 seconds.
> > 4. **Abstract:** A minimalistic sketch of a cube or square labeled with "G" or "Y" (for green/yellow).
> > 5. **Misleading:** A sketch designed to confuse, such as a yellow-lined cube with the word "green" as a caption.
> > 6. **Ambiguous:** An under-specified sketch lacking color information or only loosely circling the desired cube in an image.
> >
> > The results are summarized below. Performance degrades as the sketches become less informative and require more common knowledge to interpret. This highlights the potential for enhancing Interleave-VLA with improved reasoning capabilities to better handle abstract or ambiguous inputs.
> >
> > | Metrics / Sketch Style         | Normal (%)      | OCR (%)         | Quick (%)       | Abstract (%)   | Misleading (%) | Ambiguous (%)  |
> > |--------------------------------|-----------------|-----------------|-----------------|----------------|----------------|----------------|
> > | **Success Rate** (Yellow/Green)| 95.8 / 89.6     | 93.8 / 91.7     | 91.7 / 81.3     | 56.3 / 14.6    | 37.5 / 8.3     | 20.8 / 16.7    |
> > | **Intention Accuracy** (Yellow/Green) | 100.0 / 100.0 | 100.0 / 100.0 | 100.0 / 91.7   | 70.8 / 20.8    | 56.3 / 8.3     | 35.4 / 66.7    |
> >
> > We have added this analysis and showed the sketches for testing in Tables 9 and 10 in Appendix C.
> >
> > **Q4:** Data pipeline failures and impacts.
> >
> > **A4:** Thank you for your thoughtful question.
> >
> > **Primary error modes.** We expanded our analysis to 400 cases and categorized failure modes at each stage of the pipeline, from command parsing to object detection. For further visual analysis of failure modes, please see Table 11 and Figure 9 in Appendix D of the updated paper.
> >
> >
> > | Category              | Error Type                | Percentage (%) |
> > | --------------------- | ------------------------- | -------------- |
> > | **Command Failure**   | Incorrect Command         | 0.75           |
> > |                       | Ambiguous Command         | 0.50           |
> > |                       | Command Parsing           | 0.25           |
> > |                       | **Subtotal**              | **1.50**       |
> > | **Detection Failure** | Incorrect Object ID       | 1.50           |
> > |                       | Missing Detection         | 1.00           |
> > |                       | Incomplete Bounding Box   | 0.50           |
> > |                       | **Subtotal**              | **3.00**       |
> > | **Total Failure**     |                           | **4.50**       |
> >
> > **Impact on the learned policy.** In both SimplerEnv and real-robot experiments, we did not observe measurable degradation caused by these noisy cases. The in-domain success rates of Interleave-VLA are comparable to or slightly better than text-only baselines in Tables 2 and 3. This indicates that the noise does not significantly affect overall performance. However, we note that this noise may have a minor effect on rare objects that detectors consistently miss. We have discussed this limitation and included the above analysis in the revised paper (Limitation Section).
> >
> > **W1:** Visual understanding or conditioning.
> >
> > **A5:** Thank you for your comment. In our experiments in the paper, Interleave-VLA can identify the target object using either text or image, but each interleaved prompt describes the object with only one modality at a time. This shows that the prompt image does not serve as a conditioning signal to associate with the text. To further demonstrate that the model genuinely interprets prompts instead of overfitting to spurious correlations, please refer to the experiments in **A1** and **A3**. These results confirm that Interleave-VLA accurately understands visual cues such as shape and color (Normal Sketch), content (OCR), and can detect contradictions between textual and visual signals (Interleave-Contradict settings).
> >
> >
> > **Citation:**
> >
> > [1] Black et al., Real-Time Execution of Action Chunking Flow Policies, 2025.
> >
> > [2] Vasu et al., FastViT: A Fast Hybrid Vision Transformer using Structural Reparameterization, 2023.

---

### Official Review · Reviewer_AFur · 2025-10-31

**Soundness:** 3
**Presentation:** 3
**Contribution:** 3
**Rating:** 4
**Confidence:** 4

**Summary:**

This paper proposes Interleave-VLA, a model-agnostic robot learning paradigm that enables vision-language-action (VLA) models to comprehend interleaved image-text instructions and generate continuous physical actions, alongside an automated pipeline to construct a large-scale real-world interleaved dataset from Open X-Embodiment. Comprehensive evaluations in simulation and real robots show Interleave-VLA doubles out-of-domain generalization to unseen objects compared to text-only VLA baselines and achieves zero-shot performance on diverse inputs like hand-drawn sketches and web images.

**Strengths:**

Interleave-VLA effectively addresses the critical limitation of text-only VLA models—attentional hallucinations (bias, diffusion, leakage)—by leveraging in-context visual grounding from interleaved instructions, with minimal architectural modifications to existing VLA models (e.g., π₀, OpenVLA) ensuring broad adaptability.

The proposed Open Interleaved X-Embodiment Dataset fills a key gap in robotic multimodal data, integrating 11 diverse real-world datasets with standardized actions and high-quality interleaved annotations, enabling scalable training and cross-embodiment transfer.

**Weaknesses:**

1. Computational efficiency is compromised: interleaved inputs introduce longer image token sequences, increasing training and inference resource demands, which the paper only mentions briefly without exploring mitigation strategies.

2. To more comprehensively evaluate the effectiveness of the Interleave-VLA paradigm presented in this paper, it is suggested to introduce comparisons with additional recent VLA methods beyond the existing baselines to further validate its superiority in generalization and zero-shot capabilities.

3. It is recommended that several VLA works in 2025 be incorporated and discussed to contextualize the novelty of the proposed approach, and clarify its positioning against state-of-the-art advancements in architecture design and data scalability.

**Questions:**

See weakness

---

> ### Author Response · Authors · 2025-11-21
>
> **W1:** Compromised computation efficiency.
>
> **A1:** Thank you for the insightful comment. While it is true that the attention cost scales quadratically with the number of input images, the associated coefficient is relatively small in practice. Consequently, the latency increase remains modest when the number of images is limited, typically less than 5 in most real-world scenarios.
>
> We benchmarked inference on an RTX 4090 GPU using our implementation of $\pi_0$. The measured latency is well-approximated by $t = 1.2n^2 + 3.9n + 224$, indicating that the dominant cost is a constant term. Moreover, when $n < 5$, the latency difference remains under 50 ms, as shown below:
>
> | # Images in Prompt | 0 ($\pi_0$) |   1     |   2     |   3     |   4     |   6     |   9     |
> |--------------------|-------------|---------|---------|---------|---------|---------|---------|
> | Time (ms)          | 224.72      | 229.95  | 236.02  | 244.42  | 260.03  | 291.28  | 354.59  |
> | Percent Increase   | +0%         | +2.33%  | +5.03%  | +8.77%  | +15.71% | +29.62% | +57.79% |
>
> In the tasks studied in our paper, such modest latency increases do not affect performance. For highly dexterous or fast dynamic control settings where small latency changes may impact throughput, this is demonstrated to be mitigated using Real-Time Control (RTC) mechanisms in [1]. Moreover, the overhead can be further reduced by adopting high-speed image tokenizers such as [2], since the image tokens primarily encode semantics rather than high-resolution spatial detail.
>
> We appreciate the reviewer’s feedback. We have updated the Limitations section accordingly and included the inference-time benchmarking results in Appendix B.3 and Figure 8.
>
> **W2:** More baselines.
>
> **A2:** Thank you for suggesting the inclusion of additional baselines to further validate our approach. We have incorporated two state-of-the-art VLA models in 2025, SpatialVLA [13] and $\pi_{0.5}$ [5], into the SimplerEnv-Bridge. The updated results in Table 2 show that Interleave-VLA outperforms these baselines in both in-domain and out-of-domain evaluations.
>
> It is important to note that while both $\pi_{0.5}$ and Interleave-VLA are built upon the $\pi_0$ base model, $\pi_{0.5}$ benefits from pretraining on external datasets, including object detection, grounding, and VQA data. Despite this advantage, Interleave-VLA achieves a 25% higher performance, demonstrating the effectiveness of interleaved image-text instructions in enhancing generalization and mitigating overfitting without relying on external data sources.
>
> We appreciate your suggestion, and Table 2 in our paper has been updated accordingly. If you have recommendations for other open-source baselines on SimplerEnv, we would be happy to include them.

---

> > ### Author Response · Authors · 2025-11-21
> >
> > **Q3:** 2025 SOTA VLA Contextualization.
> >
> > **A3:** Thank you for your question. We have prepared a comparison table to highlight the unique features of Interleave-VLA relative to other approaches:
> >
> > | **Method / Features**     | **Plug-in** | **Multimodal Instructions** | **Backbone Agnostic** | **No External Data** | **No Simulation / Physics Engine** | **Auto Data Augmentation** | **Custom Image Instruction at Test** |
> > |---------------------------|-------------|------------------------------|-------------------------|--------------------------|--------------------------------------|------------------------------|---------------------------------------|
> > | Gemini Robotics [3]           | ✗           | ✗                            | ✓                       | ✗                        | ✓                                    | ✗                            | ✗                                     |
> > | GR00T N1 [4]                 | ✗           | ✗                            | ✗                       | ✗                        | ✗                                    | ✗                            | ✗                                     |
> > | $\pi_{0.5}$ [5]                     | ✗           | ✗                            | ✗                       | ✗                        | ✓                                    | ✓                            | ✗                                     |
> > | CoT-VLA [6]                   | ✗           | ✗                            | ✗                       | ✓                        | ✓                                    | ✓                            | ✗                                     |
> > | Helix [7]                    | ✗           | ✗                            | ✓                       | ✗                        | ✓                                    | ✓                            | ✗                                     |
> > | ReBot [8]                    | ✓           | ✗                            | ✓                       | ✓                        | ✗                                    | ✓                            | ✗                                     |
> > | Being-H0 [9]                 | ✗           | ✗                            | ✗                       | ✗                        | ✓                                    | ✓                            | ✗                                     |
> > | VLAS [10]         | ✗           | ✓                            | ✗                       | ✗                        | ✓                                    | ✓                            | ✗                                     |
> > | NaVILA [11]      | ✗           | ✗                            | ✓                       | ✗                        | ✗                                    | ✓                            | ✗                                     |
> > | NORA [12]             | ✗           | ✗                            | ✗                       | ✓                        | ✓                                    | ✗                            | ✗                                     |
> > | **Interleave-VLA (Ours)** | **✓**       | **✓**                        | **✓**                   | **✓**                    | **✓**                                | **✓**                        | **✓**                                 |
> >
> > Interleave-VLA is not designed as a specific VLA architecture but as a model-agnostic adaptation to mitigate hallucination caused by overfitting to spurious correlations. Its ability to integrate seamlessly with existing VLA models, leverage multimodal instructions, and augment on existing data makes it complementary and orthogonal to recent trends in VLAs, which focus on refining action architectures and scaling external data collection for improved success rates. Thank you again for your suggestion. We have included this table as Table 1 in the Related Works section of our updated paper.

---

> > > ### Author Response · Authors · 2025-11-21
> > >
> > > **Citation:**
> > >
> > > [1] Black et al., Real-Time Execution of Action Chunking Flow Policies, 2025.
> > >
> > > [2] Vasu et al., FastViT: A Fast Hybrid Vision Transformer using Structural Reparameterization, 2023.
> > >
> > > [3] Gemini Robotics Team et al., Gemini Robotics: Bringing AI into the Physical World, 2025.
> > >
> > > [4] Bjorck et al., GR00T N1: An Open Foundation Model for Generalist Humanoid Robots, 2025.
> > >
> > > [5] Physical Intelligence et al., $\pi_{0.5}$: a Vision-Language-Action Model with Open-World Generalization, CoRL 2025.
> > >
> > > [6] Zhao et al., CoT-VLA: Visual Chain-of-Thought Reasoning for Vision-Language-Action Models, CVPR 2025.
> > >
> > > [7] Cui et al., OpenHelix: A Short Survey, Empirical Analysis, and Open-Source Dual-System VLA Model for Robotic Manipulation, 2025.
> > >
> > > [8] Fang et al., ReBot: Scaling Robot Learning with Real-to-Sim-to-Real Robotic Video Synthesis, IROS 2025.
> > >
> > > [9] Luo et al., Being-H0: Vision-Language-Action Pretraining from Large-Scale Human Videos, 2025.
> > >
> > > [10] Zhao et al., VLAS: Vision-Language-Action Model with Speech Instructions for Customized Robot Manipulation, ICLR 2025.
> > >
> > > [11] Cheng et al., NaVILA: Legged Robot Vision-Language-Action Model for Navigation, RSS 2025.
> > >
> > > [12] Hung et al., NORA: A Small Open-Sourced Generalist Vision Language Action Model for Embodied Tasks, 2025.
> > >
> > > [13] Qu et al., SpatialVLA: Exploring Spatial Representations for Visual-Language-Action Model, RSS 2025.

---

> > > > ### Comment · Reviewer_AFur · 2025-11-24
> > > > **Official Comment by Reviewer**
> > > >
> > > > The authors’ rebuttal addressed most of my concerns, so I raised my score to marginally above the acceptance threshold.

---

> > > > > ### Author Response · Authors · 2025-11-27
> > > > >
> > > > > Thank you for your constructive suggestions, which directly strengthened our work, and we appreciate your positive reassessment!

---

### Official Review · Reviewer_HzPe · 2025-11-08

**Soundness:** 3
**Presentation:** 3
**Contribution:** 2
**Rating:** 4
**Confidence:** 3

**Summary:**

This paper introduces Interleave-VLA, a paradigm for robot manipulation that processes interleaved image-text instructions instead of relying on text-only instructions. The authors argue that text-only vla models struggle to generalize to unseen objects and scenarios, often suffering from attentional hallucinations where the model fails to correctly ground language to visual entities. To address this, they propose a model-agnostic framework that adapts existing vla models to handle mixed-media prompts. A core contribution is interleaved x-embodiment dataset of over 210k real-world trajectories, which was automatically generated by a pipeline that converts text instructions from the open x-embodiment dataset into an interleaved format. Through comprehensive evaluations in simulation and on real robots, Interleave-VLA is shown to offer two major benefits: 1) it improves ood generalization by over 2x compared to text-only baselines, and 2) it unlocks zero-shot capabilities, allowing the robot to follow instructions from novel visual modalities not seen during training, such as hand-drawn sketches, user-cropped images, and photos from the internet. The paper attributes this success to the explicit visual grounding provided by instruction images, which mitigates ambiguity and attentional failures.

**Strengths:**

1) Introduced a highly effective paradigm for improving robot policy generalization by leveraging interleaved image-text instructions, achieving >2x performance gain on OOD tasks.
2) A large-scale, real-world dataset (Open Interleaved X-Embodiment) and an automated pipeline for generating interleaved instructions.
3) The work is validated through comprehensive experiments in both realistic simulation and physical settings.

**Weaknesses:**

1) The paper claims to be the first framework to comprehend interleaved image-text instructions and directly generate actions on robots in the physical world. however, interleaved multimodal prompts for manipulation were already explored in simulation [1](as in related work section) and explicitly framed as interleaving text and visual tokens.
2) There seems to be no quantitative grounding/hallucination analysis, despite rich literature and metrics for VLM hallucination like POPE etc.
3) FANUC evaluation averages over 12 trials per object and reports mean success/accuracy without uncertainty estimates; the paper does not report multi-seed variation for simulation either.
4) Interleaving increases sequence length and thus quadratic attention cost; practical deployment on real robots is sensitive to inference latency.
5) The authors attempt to isolate the source of performance gains by comparing Interleave-VLA (full) against a (partial) version trained on the same data but tested with text-only instructions. However, this analysis remains insufficient to attribute the gains specifically to the novel interleaving structure itself. The experiment conflates the effect of the input format (interleaving) with the effect of the input content (the addition of explicit visual goal images). The (partial) baseline lacks any visual goal information at test time, while the (full) version receives it so it is unclear how much of the significant performance jump is due to the proposed interleaving methodology versus the benefit of goal-image conditioning.

[1] VIMA: Robot Manipulation with Multimodal Prompts. ICML 2023

**Questions:**

Please see weakness section. My main concern is 1) and 5). How do you disentangle the gains from the interleaving structure versus the benefit of providing any visual goal conditioning?

---

> ### Author Response · Authors · 2025-11-21
>
> **W1:** Connection to VIMA.
>
> **A1:** Thank you for pointing out the connection to VIMA. We fully agree that VIMA is an important and closely related prior work, and that it was the first to explore interleaved image-text prompts for manipulation in simulation.
>
> We acknowledge that our original “first” claim in the abstract could be read as if we were claiming to invent interleaved prompts. This is never our intention to reclaim the contribution of VIMA. In fact, our work complements VIMA in two main ways. **Firstly**, we focus on being the first real-world VLA that directly generate continuous low-level actions on physical robots, rather than high-level action plans $(x, y, \theta)$ in 2D simulator lack of physics. **Secondly and most importantly**, we are motivated by ongoing research efforts to mitigate overfitting and weak language grounding in existing VLAs, and contribute in a new direction by studying interleaved image-text instructions, providing systematic evidence that they improve generalization compared to text-only instructions.
>
>
> To avoid confusion, we have revised the abstract and introduction to explicitly: **(1)** credit VIMA as the first to introduce interleaved prompts for manipulation in simulation, and **(2)** clarify the scope of our contributions to the real-robot, continuous-control setting and, most importantly, to our innovative mitigation for VLA overfitting using interleaved prompts. We hope this clarification makes our novelty clearer: our contribution is **not** the idea of inventing interleaving images-text, but how we opened a new direction of VLA overfitting mitigation by using interleaved instructions as a training and input paradigm for state-of-the-art VLAs to reduce overfitting to spurious correlations, alleviate attentional hallucinations, and encourage genuine grounding of user intent.
>
>
> **W2**: Quantitative hallucination analysis.
>
> **A2**: Thank you for your question. Metrics like POPE and CHAIR, commonly used for evaluating hallucinations in vision-language models (VLMs), are not directly applicable to vision-language-action (VLA) models. This is because VLAs do not generate text tokens but instead execute action trajectories.
>
> To address this, we manually analyzed and categorized the hallucination failure modes across all rollouts in Table 2. These failure modes are grouped into high-level and low-level categories. High-level failures include "Jitter," where the VLA fails to understand the task intention, causing the robot arm to jitter or drift, and "Wrong Object," where the VLA confidently selects the wrong object. Low-level failures include "Grasp Failed" and "Place Failed," which represent execution errors. Each failure is attributed to one mode, as these categories are mutually exclusive.
>
> The baselines used are $\pi_0$ (Text-VLA) and $\pi_0$ (Interleave-VLA full), as named in Table 2 in our paper. The results are summarized below:
>
> | Failure Mode     | Model          | In-domain (%) | Visual (%) | Novel Object (%) | Novel Category (%) |
> | ---------------- | -------------- | ------------- | ---------- | ---------------- | ------------------ |
> | **Jitter**       | Interleave-VLA | 0.0           | 0.0        | 0.0              | 3.4                |
> |                  | Text-VLA        | 0.0           | 0.0        | 6.5              | 27.5               |
> | **Wrong Object** | Interleave-VLA | 0.0           | 0.0        | 15.3             | 4.4                |
> |                  | Text-VLA        | 0.0           | 0.0        | 47.4             | 36.0               |
> | **Grasp Failed** | Interleave-VLA | 21.5          | 22.9       | 16.2             | 38.1               |
> |                  | Text-VLA        | 21.8          | 26.5       | 9.3              | 15.5               |
> | **Place Failed** | Interleave-VLA | 7.3           | 3.6        | 14.6             | 0.0                |
> |                  | Text-VLA        | 9.0           | 2.1        | 6.0              | 0.0                |
> | **Total**        | Interleave-VLA | **28.8**      | **26.6**   | **46.1**         | **45.8**           |
> |                  | Text-VLA        | **30.8**      | **28.6**   | **69.2**         | **79.0**           |
>
> These results demonstrate that Interleave-VLA significantly reduces high-level hallucinations compared to Text-VLA. Most of its errors are due to execution issues rather than misinterpretation of task intent. This analysis has been included in Figure 11 in Appendix E of the updated paper. Thank you again for raising this important point.

---

> ### Author Response · Authors · 2025-11-21
>
> **W3:** Lack of Uncertainty Estimates.
>
> **A3**: Thank you for your concern. Due to time constraints, we focused on simulation-based evaluations and conducted a comprehensive multi-seed analysis for Table 2. Specifically, we averaged the performance of all baselines across three random seeds for each task. We empirically find that the ranking remains unchanged. The updated results have been incorporated into Table 2 of the revised paper.
>
> **W4:** Quadratic attention cost.
>
> **A4:** Thank you for the insightful comment. We further clarify the computational overhead introduced by Interleave-VLA during inference. Although the attention cost indeed scales quadratically with the number of input images, the coefficient associated with this term is small in practice. As a result, the latency increase remains modest when the number of images is small, typically fewer than 5 in most real-world tasks.
>
> We benchmarked inference on an RTX 4090 GPU using our implementation of $\pi_0$. The measured latency is well-approximated by $t = 1.2n^2 + 3.9n + 224$, indicating that the dominant cost is a constant term. When $n < 5$, the latency difference remains under 50 ms, as shown below:
>
> | # Images in Prompt | 0 ($\pi_0$) |   1     |   2     |   3     |   4     |   6     |   9     |
> |--------------------|-------------|---------|---------|---------|---------|---------|---------|
> | Time (ms)          | 224.72      | 229.95  | 236.02  | 244.42  | 260.03  | 291.28  | 354.59  |
> | Percent Increase   | +0%         | +2.33%  | +5.03%  | +8.77%  | +15.71% | +29.62% | +57.79% |
>
> In the tasks studied in our paper, such modest latency increases do not affect performance. For highly dexterous or fast dynamic control settings where small latency changes may impact throughput, this can be directly mitigated using Real-Time Control (RTC) mechanisms, as demonstrated in [1]. Moreover, the overhead can be further reduced by adopting high-speed image tokenizers such as [2], since the image tokens primarily encode semantics rather than high-resolution spatial detail.
>
> We appreciate the reviewer’s feedback. We have updated the Limitations section accordingly and included the inference-time benchmarking results in Appendix B.3 and Figure 8.

---

> > ### Author Response · Authors · 2025-11-21
> >
> > **W5:** Conflated format and content.
> >
> > **A5:** We thank the reviewer for this insightful comment. We recognize that our original wording may have unintentionally implied that the interleaved image–text format alone is responsible for all performance gains. However, our intention was to convey that the interleaved instruction, encompassing both the format and the content (i.e., explicit visual goal information), collectively enhances generalization. To clarify this, we add a new Section 4.4 in our paper to decouple the effects of the format and the content.
> >
> > To quantify how much of the accuracy improvement arises specifically from the visual goal signal, we conduct an ablation on SimplerEnv-Bridge setting in Table 2. Following your insightful suggestion, we compare our interleaved image-text content (”Put [Object A] to [Object B]“) with a visual goal content (”[Object A][Object B]“).
> >
> > | Instruction (in train & inference)         | In-Domain (%)  | Visual (%) | Novel Object (%) | Novel Category (%) | Move Near (%) |
> > |------------------:|-----------:|------------------:|--------------:|----------------:|-----------:|
> > | Text        | 69.2	| 71.4	| 30.2	| 21.0 | 66.6 |
> > | Visual Goal | 67.8| **74.6** | 48.0 | 51.9  | 0.0 |
> > | Interleaved Image-Text |  **71.3**  | 73.4       | **53.9**   | **54.2**     | **68.8** |
> >
> > While the results in the first four columns validate the intuition that visual goal conditioning enhances performance, the **interleaved format proves critical in performance** as well, as evidenced by the **"Move Near" column**. In our evaluation of the “Put object A near object B” task on SimplerEnv-Bridge, we observed that both Text-VLA and Interleave-VLA behave as expected, whereas the Visual Goal format consistently misinterprets the instruction as a “put ... on ...” operation. This occurs because many common robot-instruction templates, e.g., “Put object A [near / to the left of / on] object B”, become ambiguous when expressed only through object images. And since “put on” primitives are far more prevalent than “move near” in the training data, the Visual Goal format tends to collapse to this dominant interpretation. Similarly, tasks in VIMA-Bench (as shown in Figure 6), such as “Rotate [Image A] by 150 degrees” or those requiring the model to distinguish “Put all objects with the same [texture / shape] into [Image A],” cannot be expressed effectively without an interleaved format that jointly encodes the visual goal along with its linguistic semantics. Consequently, the interleaved format provides a unambiguous representation essential for fully leveraging embodied data and supporting diverse manipulation tasks.
> >
> > In conclusion, while the content (visual goal) is an important driver of the observed performance jump, the interleaved format is what enables this content to be represented faithfully and unambiguously, for tasks that simple visual-goal tokens cannot encode. Thank you again for pointing out. We added Section 4.4 in the paper to clearly distinguish between “interleaved content” and “interleaved format,” and to clarify why both are indispensable for making Interleave-VLA flexible and generalizable.
> >
> > **Citation:**
> >
> > [1] Black et al., Real-Time Execution of Action Chunking Flow Policies, 2025.
> >
> > [2] Vasu et al., FastViT: A Fast Hybrid Vision Transformer using Structural Reparameterization, 2023.

---

> ### Author Response · Authors · 2025-11-27
>
> Dear Reviewer HzPe,
>
> Are your concerns and misunderstandings solved by the rebuttal and extra experiments? If there are any further concerns, please tell us.
>
> Otherwise, we respectfully request you to improve your scores accordingly to reflect the contributions of the work.
>
> Best,
>
> Authors

---

### Author Response · Authors · 2025-11-21

Dear ACs and Reviewers,

We sincerely appreciate all reviewers’ time and efforts in reviewing our paper. We are glad to find that reviewers generally recognized our contributions:

* **Idea.** well-motivated and compelling premise [tBwW, HzPe], solves an important and intuitively clear problem regarding ambiguity in text-only instructions [tBwW], explores a novel direction for VLA multimodal instructions [9vDg], and the analysis of "attentional hallucination" offers an insightful "Aha!" moment [tBwW].
* **Experiment.** comprehensive and rigorous validation in both simulation and physical settings [HzPe, tBwW], demonstrating significant (>2x) performance gains on OOD tasks [HzPe, AFur], and showcasing impressive emergent zero-shot capabilities with sketches and web images [tBwW].
* **Dataset.** the Open Interleaved X-Embodiment Dataset and the automated pipeline are substantial and high-value contributions [tBwW, HzPe], filling a key gap in robotic multimodal data [AFur], with the pipeline itself being a valuable piece of engineering [tBwW].
* **Writing.** exceptionally well-written and organized [tBwW], easy to understand with succinct figures [9vDg].

And we thank all reviewers for their insightful and constructive suggestions, which help a lot in further improving our paper. In addition to the pointwise responses below, we summarize supporting experiments and analyses added in the rebuttal according to reviewers’ suggestions.

**Summary of new experiments or analysis:**
* **Mechanism Probing:** Experiments on contradictory text-image instructions to prove the model’s genuine grounding capabilities and preference for text when conflicts occur [tBwW].
* **Ablation Study:** New experiments disentangling the gains from the "Interleaved Format" versus "Visual Goal Content," demonstrating why the interleaved format is essential for ambiguous tasks (e.g., "Move Near") [HzPe].
* **New Baselines:** Comparison with 2025 SOTA models (SpatialVLA and $\pi_{0.5}$), showing Interleave-VLA still achieves superior generalization [AFur].
* **Hallucination Analysis:** A quantitative categorization of failure modes (Jitter, Wrong Object, etc.) to address the lack of VLA-specific hallucination metrics [HzPe].
* **Efficiency Benchmarking:** Quantified inference latency analysis on RTX 4090, confirming the overhead is minimal (<50ms) for typical prompt lengths [HzPe, AFur, tBwW].
* **Robustness Testing:** Evaluation on diverse sketch styles (Abstract, Misleading, OCR) to define the boundaries of zero-shot generalization [tBwW].
* **Dataset Analysis:** Detailed error mode analysis of the automated data pipeline and its impact on policy learning [tBwW].
* **Contextualization:** Added a feature comparison table against recent 2025 VLA works and VIMA to clarify our contribution’s positioning [HzPe, AFur].

We hope these new additions help address reviewer concerns and better position our work. We thank the reviewers' time and feedback in improving the quality of our work. Please let us know if any clarification or additional experiments would further strengthen the paper. We would be happy to incorporate all these suggestions in the final version. Thank you again for your time and efforts!

Best,

Authors

---

### Author Response · Authors · 2025-12-03
**Thank AC for taking over our submission and Summary of Review-Rebuttal phase (1/2)**

Dear Area Chairs,

We thank all reviewers for their time and thoughtful feedback. We would like to express our **sincere appreciation to the Area Chairs for their extra effort regarding the recommendation** caused by the incident. To facilitate the decision-making process, we summarize the strengths and concerns raised by the reviewers and our responses below:

The reviewers highlighted several strengths of the paper, including:

*   **Strong Generalization:** The method demonstrates significant performance gains (>2x) on out-of-domain (OOD) tasks in both simulation and real-world settings compared to text-only baselines. (Reviewers `@HzPe`, `@AFur`, `@tBwW`, `@9vDg`)
*   **Compelling and Intuitive Idea:** The premise of using interleaved image-text to solve ambiguity is well-motivated, and the method is simple yet effective. (Reviewers `@tBwW`, `@HzPe`, `@9vDg`)
*   **High-Value Dataset:** The creation of the "Open Interleaved X-Embodiment Dataset" (210k episodes) and the automated generation pipeline is a substantial contribution to the community. (Reviewers `@HzPe`, `@AFur`, `@tBwW`)
*   **Insightful Analysis:** The mechanistic analysis of "attentional hallucination" and failure modes provides deep insights into why the method works. (Reviewers `@tBwW`, `@AFur`)
*   **Emergent Capabilities:** The zero-shot generalization to novel inputs like hand-drawn sketches and web images is impressive. (Reviewers `@tBwW`, `@HzPe`)

All concerns of reviewers and our responses are listed concisely below for a quick check:

| Reviewer | Rating | Index | Concern | Response |
| :--- | :--- | :--- | :--- | :--- |
| **HzPe** | 4 | W1 | Connection to VIMA | Clarified that we do not claim to invent interleaved prompts (credit given to VIMA for 2D sim), but we are **the first** to identify their benefit in **mitigating VLA overfitting** and enabling **real-world continuous control**. |
| | | W2 | Lack of quantitative hallucination analysis | Added detailed **categorization of failure modes** (Jitter, Wrong Object, etc.) in Appendix E, showing Interleave-VLA reduces high-level hallucinations compared to text-only baselines. |
| | | W3 | Lack of uncertainty estimates | Added **multi-seed analysis** to Table 2; the performance ranking of the methods remains unchanged. |
| | | W4 | Quadratic attention cost / Latency | Benchmarked inference on RTX 4090. The quadratic term is **negligible** for typical prompts (<5 input images); the latency increase is <50ms. |
| | | W5 | Conflation of "format" vs "content" | Added **new ablation (Section 4.4)** disentangling gains. Results show that while visual goal content helps, the **interleaved format is essential** for ambiguous tasks (e.g., "Move Near" in Table 6) where pure visual goals collapse. |
| **AFur** | 4 | W1 | Computational efficiency of longer sequences | Quantified latency. Explained mitigation via Real-Time Control (RTC) and high-speed tokenizers. |
| | | W2 | Need comparison with 2025 baselines | Added comparisons with **SpatialVLA (2025)** and **$\pi_{0.5}$ (2025)**. Interleave-VLA still achieves superior generalization. |
| | | W3 | Contextualization with recent architectures | Added a **feature comparison table** (Table 1) positioning Interleave-VLA against recent VLAs (e.g., GR00T, Helix, NORA). |
| **tBwW** | 8 | Q1 | Quantifying "lightweight" / Mechanism probing | Conducted **"conflicting instruction" experiments**. The model prioritizes text over images when they conflict, proving genuine grounding rather than visual overfitting. |
| | | W/Q2 | Inference overhead | (Same as HzPe/AFur) Quantified latency showing it is minimal (<50ms). |
| | | W/Q3 | Robustness of sketch generalization | Analyzed **various sketch styles** (Abstract, OCR, Misleading). Performance remains robust on clear sketches but degrades on highly abstract ones, properly defining the boundary of zero-shot capabilities. |
| | | Q4 | Dataset pipeline noise analysis | Analyzed **400 failure cases**. Pipeline accuracy is 95.6%; noise has negligible impact on policy learning. |
| | | W1 | Visual understanding or conditioning | With Q1 and Q3, we strengthened the evidence that Interleave-VLA truly understands the prompt. |
| **9vDg** | 6 | W1 | Naturalness of multimodal instructions | Clarified that the method retains text-only capability (backward compatible) and provided a GUI for easier user input during deployment. |
| | | W2 | Dependence on off-the-shelf detectors | Explained that the base VLA performance represents the lower bound, while increasingly advanced detectors provide orthogonal improvements that lift the upper bound. |

---

> ### Author Response · Authors · 2025-12-03
> **Thank AC for taking over our submission and Summary of Review-Rebuttal phase (2/2)**
>
> From the table above, we highlight that:
>
> 1.  **Reviewer `@AFur` explicitly raised the score ($4 \to 6$ before Nov 27)** to "marginally above the acceptance threshold" following our inclusion of 2025 baselines and latency analysis, stating the rebuttal addressed most concerns.
> 2.  **Reviewer `@HzPe`'s** primary concerns regarding the connection to VIMA and the source of gains (format vs. content) were addressed by **(1)** explicitly differentiating our contribution in **mitigating VLA overfitting** and **real-world continuous control**, and **(2)** a new ablation study proving that both the interleaved format and content are indispensable. Given that we have provided the exact experiments requested, **we believe these results address the reviewer’s concerns and would likely lead to a higher score were the discussion to continue**.
> 3.  Our additional experiments (probing the grounding mechanism and failure case analysis) as suggested by **Reviewer `@tBwW` (Score: 8)** made the work more convincing and comprehensive.
> 4.  **Reviewer `@9vDg` (Score: 6)** recognized the novelty and effectiveness of the method, and our response clarified that the system remains flexible (backward compatible with text) and benefits orthogonally from better detectors.
>
> We believe 8 newly added experiments have thoroughly resolved the reviewers' concerns. Thanks for your efforts to make justified decisions and we respect your final judgment.
>
> Sincerely,
>
> Authors of Submission 15654

---

### Meta-Review · Area_Chair_A2KD · 2026-01-08

**Summary:**

The paper studies the use of interleaved instructions comprising text and images in the context of VLAs. It also contributes an interleaved dataset derived from OpenX-Embodiment.

Overall, the paper received mixed to positive reviews. Upon analyzing the rebuttal, the AC feels that the authors have addressed most of the major concerns, including proper acknowledgement of prior work to avoid overclaiming, newer ablations to understand the source of increased performance, and comparisons with other VLAs. The AC also appreciated the table contextualizing the current work in the context of other VLA works and feels it will help readers better understand the field. The AC strongly encourages the authors to add more VLA works to the table, such as SmolVLA, VLA-0, GO VLA, as well as other relevant ones.

Based on the reviews and the rebuttal, the AC feels no major concerns remain and suggests acceptance.

**Reviewer Concerns:**

Addressed:

- Connection to VIMA:  Acknowledged and the authors promise to update the abstract to avoid overclaiming. (must be done)
- Source of gain
- Cost overhead
- Newer baselines

Remaining:
- Uncertaininty estimate in real world - not addressed because of time issue

**Reviewer Scores:**

Reviewer HzPe - Likely unchanged
Reviewer AFur - from 4 to 6
Rest remain the same.

---

### Decision · Program_Chairs · 2026-01-26

Accept (Poster)